# The NMDA receptor regulates competition of epithelial cells in the *Drosophila* wing

Agnes R. Banreti[1,2✉] & Pascal Meier [1✉]

Cell competition is an emerging principle that eliminates suboptimal or potentially dangerous cells. For 'unfit' cells to be detected, their competitive status needs to be compared to the collective fitness of cells within a tissue. Here we report that the NMDA receptor controls cell competition of epithelial cells and Myc supercompetitors in the *Drosophila* wing disc. While clonal depletion of the NMDA receptor subunit NR2 results in their rapid elimination via the TNF/Eiger>JNK signalling pathway, local over-expression of NR2 causes NR2 cells to acquire supercompetitor-like behaviour that enables them to overtake the tissue through clonal expansion that causes, but also relies on, the killing of surrounding cells. Consistently, NR2 is utilised by Myc clones to provide them with supercompetitor status. Mechanistically, we find that the JNK>PDK signalling axis in 'loser' cells reprograms their metabolism, driving them to produce and transfer lactate to winners. Preventing lactate transfer from losers to winners abrogates NMDAR-mediated cell competition. Our findings demonstrate a functional repurposing of NMDAR in the surveillance of tissue fitness.

[1] The Breast Cancer Now Toby Robins Research Centre, The Institute of Cancer Research, London, UK. [2] Université Côte D'Azur, CNRS, Inserm, Institut de Biologie Valrose, Nice, France. ✉email: abanreti@unice.fr; pmeier@icr.ac.uk

Cell competition is an evolutionary conserved quality control process, which ensures that suboptimal, but otherwise viable, cells do not accumulate during development and aging[1]. How relative fitness disparities are measured across groups of cells, and how the decision is taken whether a particular cell will persist in the tissue ('winner cell') or is killed ('loser cell') is not completely understood. This is an important issue as competitive behaviour can be exploited by cancer cells[1].

Various types of cell competition exist[2]. While structural cell competition is triggered upon loss of cellular adhesion or changes in epithelial apico-basal polarity, metabolic cell competition occurs in response to alterations in cellular metabolic states. Growth signalling pathways involved in metabolic cell competition seem to funnel through Myc, which functions as an essential signalling hub in many types of cancers. Myc regulates expression of components that control proliferation, cell death, differentiation, and central metabolic pathways. Particularly, acute changes in cellular metabolism appear to be critical for the winner phenotype during Myc supercompetition in Drosophila[3], where robustly growing Myc-expressing cells are able to not only outgrow but also actively trigger the elimination of nearby wild-type cells from the tissue.

Recent in vivo data demonstrate that some tumours can uptake lactate and preferentially utilize it over glucose to fuel tricarboxylic acid (TCA) cycle and sustain tumour metabolism[4]. Moreover, the growth-promoting effect of stromal cells is impaired by glycolytic inhibition, suggesting that the stroma provides nutritional support to malignant cells by transferring lactate from cancer-associated fibroblasts (CAFs) to cancer cells[5,6]. Such energy transfer from glycolytic stromal cells to epithelial cancer cells closely resembles the physiological processes of metabolic cooperativity, such as in 'neuron-astrocyte metabolic coupling' in the brain, and the 'lactate shuttle' in the skeletal muscle[7,8]. Activation of glycolysis in astrocytes and MCT-mediated transfer of lactate to neurons supports neuron mitochondrial oxidative phosphorylation and energy demand[9]. These observations raise the intriguing possibility that lactate serves as fuel to complement glucose metabolism during cell competition.

We report here that the NMDA receptor controls the competitiveness of epithelial cells in the Drosophila wing discs. While tissue-wide depletion of NR2 has no effect on cell viability and growth, clonal depletion of NR2 results in their rapid elimination via the TNF>JNK signalling pathway. Conversely, local over-expression of NR2 causes NR2-overexpressing cells to acquire supercompetitor-like behaviour that enables them to overtake the tissue. These data indicate that relative levels of NR2 underpins cell competitive behaviour in the wing epithelia. Moreover, we find that Myc-induced supercompetition also depends on upregulation of NMDAR. Genetic depletion of NR2 abrogates Myc-induced supercompetition. Mechanistically, we find that the JNK>PDK signalling axis in 'loser' cells (lower NMDAR) results in phosphorylation and inactivation of PDH, the enzyme that converts pyruvate to Acetyl-CoA to fuel the TCA in the mitochondria. In such loser cells, phospho-dependent inactivation of PDH causes mitochondrial shutdown and metabolic reprogramming, thus loser cells produce and secrete lactate to winners. Preventing lactate transfer from losers to winners removes fitness disparities and abrogates NMDAR-mediated cell competition. Together our data are consistent with the notion that NMDAR underpins cell competition and that targeting NMDAR converts Myc supercompetitor clones into superlosers.

## Results

### NR2 drives cell competition.
In Drosophila, polarity-deficient mutant cells for discs large 1 (dlg1) are eliminated by wild-type

neighbours through cell competition[10]. dlg1 is the highly conserved homologue of mammalian PSD-95 and SAP97. In mammals, PSD-95 and SAP97 directly bind to NR2B[11,12], a subunit of the N-methyl-D-aspartate receptor (NMDAR). We, therefore, investigated whether NMDAR takes part in cell competition. While mammals encode seven different NMDAR subunits, Drosophila encodes only two NMDAR subunits (NR1 and NR2) (Fig. 1a), which simplifies their study. Consistent with previous reports[13,14], we find that Drosophila NR2 is expressed in the central nervous system, imaginal eye and wing discs as well as salivary gland and fat body (Supplementary Fig. 1a–c)[14–16]. To study the role of NR2 in cell competition in wing discs, we generated mosaic tissues of two clonal populations. This confronts wild-type cells (WT) with clones of cells in which the gene-of-interest (goi) is depleted by RNAi (marked by GFP (green fluorescent protein)) (Fig. 1b, left panel). We also, generated homotypic settings in which the goi is depleted tissue-wide, and where we created GFP-marked 'non-competitive clones' to evaluate intrinsic competition (Fig. 1b, right panel). Comparison of clonal occupancy in hetero- versus homotypic genetic backgrounds of age-matched larvae allows the exclusion of genes that compromise cell viability in general. Interestingly, clonal knockdown of NR2 (subsequently referred to as NR2 clones) using five different RNAi constructs (Fig. 1a, c, d) resulted in the loss of NR2 clones (Fig. 1c, d). Likewise, and as previously demonstrated[17,18], clonal knockdown of dlg1 or scribble (scrib) resulted in their elimination (Fig. 1d and Supplementary Fig. 2a). In contrast, clonal depletion of LacZ, which served as RNAi control, had no effect (Fig. 1c, d). Importantly, RNAi-mediated depletion of NR2 or dlg1 had no effect on clonal occupancy under homotypic condition, such as upon tissue-wide NR2 depletion using nubbin-Gal4 (Fig. 1e, f, and Supplementary Fig. 2b) or hedgehog (hh-Gal4) (Supplementary Fig. 2c) that drive expression of the RNAi constructs in the entire wing pouch or posterior compartment of the Drosophila wing imaginal disc, respectively. The observation that NR2-depleted cells are lost from the tissue when surrounded by WT cells, but are present under homotypic conditions, suggests that clonal depletion of NR2 triggers competitive interactions, resulting in the loss of otherwise viable cells.

Homotypic clonal analysis demonstrated that the intrinsic growth rate of NR2 clones is equivalent to the one of control LacZ cells (Fig. 1g), highlighting that NR2 depletion does not impair cell viability or growth in general. Further, treatment with AP5 ((2R)-amino-5-phosphonopentanoate), a selective inhibitor of NR2[19], suppressed the loss of NR2 clones in a heterotypic genetic background (losers among winners) (Supplementary Fig. 3a, b), phenocopying a homotypic setting. This illustrates that the competitive behaviour between NR2-losers and WT-winners is due to a relative difference in NR2 activity among competing clones. AP5-medited global inhibition of NR2 thereby seems to eradicate the fitness disparity among competing clones.

Next, we examined the consequence of clonal over-expression of NR2. As shown in Fig. 1h, i, NR2 over-expressing clones overgrew at the expense of wild-type surrounding cells. This suggests that elevated expression of NR2 causes NR2 cells to acquire supercompetitor-like behaviour that enables them to overtake the tissue. Of note, clonal over-expression of NR2 did not alter developmental timing, organ or larval size (Fig. 1j). Together, our data indicate that relative levels of NR2 underpins cell competitive behaviour in the wing epithelia.

### NR2 loser clones are eliminated by TNF > JNK-mediated apoptosis.
To study the elimination process of NR2-depleted cells, we examined the possible involvement of the Grindelwald>JNK signalling axis[20]. We noticed intense staining of

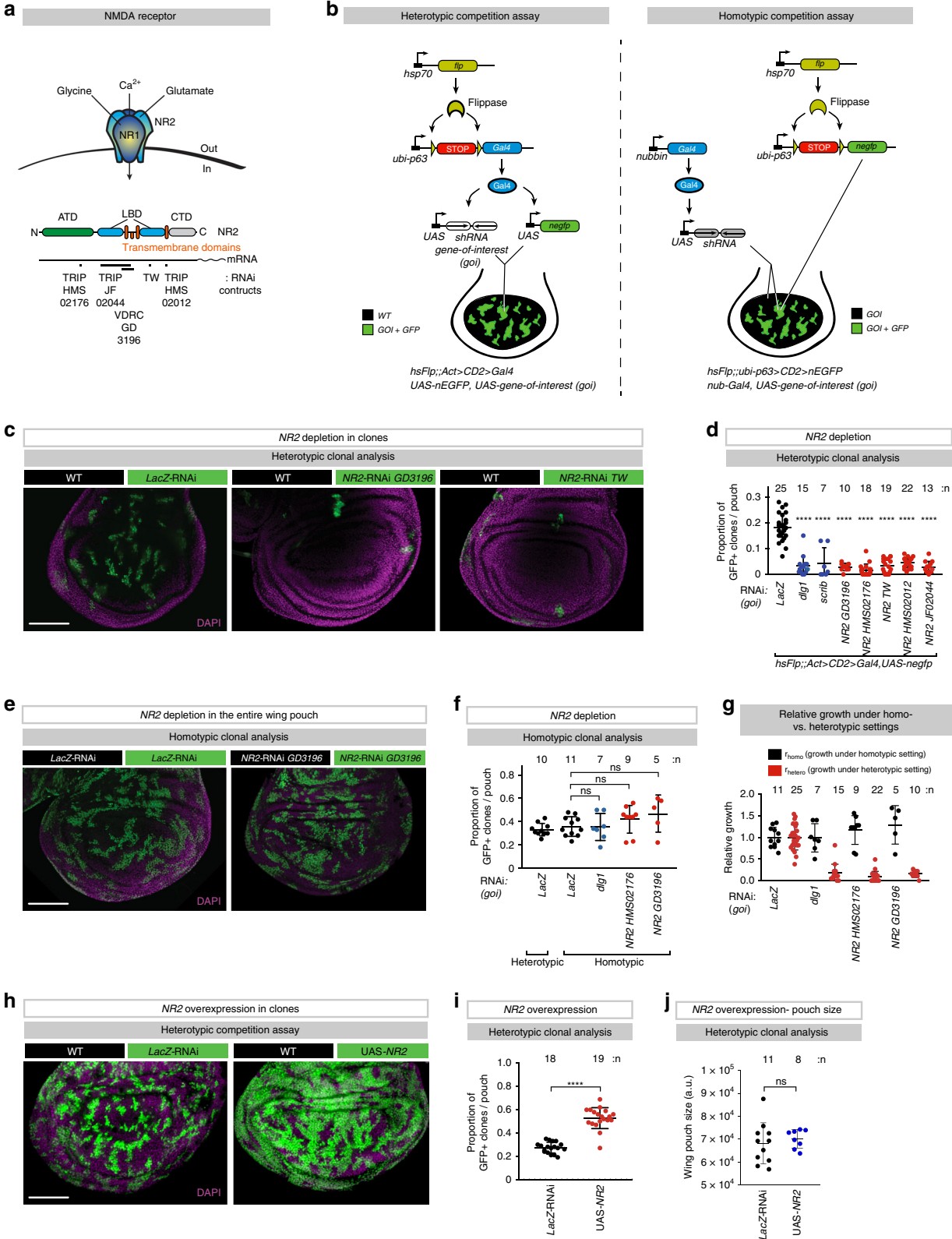

activated JNK [p-JNK] (Fig. 2a), induction of the JNK activity reporter *PucLacZ* (Supplementary Fig. 4a) and high expression level of the JNK target gene *MMP1* in and around *NR2* clones (Fig. 2b), suggesting an involvement of JNK signalling. Consistently, while NR2-depleted cells were readily eliminated, such clones survived upon simultaneous, clonal depletion of the *Drosophila* TNF-receptor superfamily member *grindelwald* (*grnd*)

(*grnd*-RNAi), *hemipterous* (*hep*) (*hep*-RNAi) or inactivation of *basket* (*bsk*) (*UAS-bsk-dominant negative (DN)*). This demonstrates that *NR2*-depleted clones are eliminated in a grnd- and JNK-dependent manner (Fig. 2c). JNK signalling in *NR2*-depleted clones ultimately resulted in caspase-mediated cell death because expression of the caspase inhibitors p35 and DIAP1 suppressed the elimination of *NR2*-depleted clones (Fig. 2c). Consistently, we

**Fig. 1 Clonal depletion of NR2 results in the elimination of otherwise viable cells. a** The *N*-methyl-ᴅ-aspartate receptor (NMDAR) is a hetero-tetrameric receptor consisting of two NR1 subunits and two NR2 subunits. Linear representation of the modular amino-terminal domain (ATD, green), ligand-binding domain (LBD, blue), transmembrane domains (red) and C-terminal domain (grey). Indicated are the names and position of the respective *NR2*-RNAi constructs used in this study. **b** Schematic representation of the genetic systems used to study cell competition in the *Drosophila* wing pouch. Left panel: Heterotypic genetic system with GFP marking loser clones. Right panel: homotypic genetic system with GFP marking non-competitive clones. **c** Heterotypic clonal analysis. The indicated genes-of-interest (*goi*) were knocked down in GFP-marked clones as depicted in a (left panel, see Methods for details). Clones are marked by GFP (green). Specific genotypes of the discs shown in these panels, and all subsequent panels, can be found in Supplementary Table 1. Scale bar 100 μm. Experiments were repeated four independent times. **d** Quantification of the heterotypic competition assay. Diagram shows the average occupancy of the indicated RNAi clones per wing pouch. **e** Homotypic clonal analysis. The indicated genes-of-interest were knocked down throughout the wing pouch as described in a (right panel). GFP-marked clones represent non-competitive clones. Scale bar 100 μm. **f** Quantification of the homotypic competition assay. Diagram shows the average occupancy of non-competitive clones in the indicated wing pouches where the respective gene-of-interest were knocked down. **g** Relative rates of growth under hetero- versus homotypic clonal settings. **h** Heterotypic clonal analysis. NR2 was over-expressed in GFP-marked (green) clones. Specific genotypes of the discs shown in these panels, and all subsequent panels, can be found in Supplementary Table 1. Scale bar 100 μm. Experiments were repeated four independent times. **i** Quantification of the heterotypic competition assay. Diagram shows the average occupancy of the indicated clones per wing pouch. **j** Quantification of the average size of wing pouches of the indicated genotypes. Error bars represent average occupancy of the indicated RNAi clones per wing pouch ± SD. ****$P < 0.0001$, ***$P < 0.001$, **$P < 0.01$, *$P < 0.1$ by Mann–Whitney two-tailed nonparametric *U*-test. (**d:** *NR2*-RNAi GD3196 vs *dlg1*-RNAi (ns) *P* value: 0.5588, *NR2*-RNAi HMS02176 vs *dlg1*-RNAi (ns) *P* value: 0.0719, *NR2 TW* vs *dlg1*-RNAi (ns) *P* value: 0.7751; **j:** *UAS-NR2* vs. *LacZ*-RNAi (ns) *P* value: 0.31). *n* depicts the number of wing discs. Experiments were repeated three independent times, unless stated otherwise. See Supplementary Table 1 for genotypes.

observed cells positive for cleaved caspase staining (Supplementary Fig. 4b). Clonal depletion of *NR2* (RNAi) sometimes lead to distortions of the disc, which most likely was due to the large amount of cell death (loser clone elimination) that occurs in these discs.

**Metabolic reprogramming and lactate production by losers**. We noticed that *NR2*-depleted clones appeared to have mitochondria that were less active and smaller in size (Fig. 3a, b). To explore this further, we investigated whether JNK signalling might influence mitochondrial function in *NR2*-depleted clones. Intriguingly, we noticed prominent JNK-dependent phosphorylation of Pyruvate Dehydrogenase (PDH), a key enzyme that catalyses the conversion of pyruvate into acetyl-CoA to be consumed in mitochondria under heterotypic (Fig. 3c and Supplementary Fig. 5a–d) but not homotypic settings (Supplementary Fig. 5e). Previous reports indicated that JNK signalling can lead to activation of the mitochondrial Pyruvate Dehydrogenase Kinase (PDK), which in turn can phosphorylate and inactivate PDH[21]. Upon phosphorylation of PDH, pyruvate is no longer converted to acetyl-CoA and therefore no longer available for the TCA in mitochondria, leading to metabolic reprogramming[22]. To test the importance of PDK-mediated phosphorylation of PDH (p-PDH) for the elimination of *NR2*-RNAi loser clones we co-depleted PDK in such clones. Interestingly, co-depletion of PDK fully rescued the elimination of *NR2* loser clones (Fig. 3d, e) and abrogated the appearance of cleaved DCP1 positive cells (Supplementary Fig. 5f). Likewise, Gal4-driven expression of PDH in *NR2* loser clones blocked their elimination (Fig. 3e and Supplementary Fig. 6a). In both settings, surviving *NR2*-depleted clones were negative for anti-p-PDH staining (Fig. 3d and Supplementary Fig. 6a). Local over-expression of PDH abrogated JNK signalling (Supplementary Fig. 6b), highlighting the presence of a feedback regulatory loop. Together, these data indicate that PDK-mediated phosphorylation and inactivation of PDH contributes to the death of *NR2*-depleted loser clones.

**Loser cells transfer their lactate to winners**. Since inactivation of PDH results in aerobic glycolysis, we assessed whether metabolic reprogramming of *NR2*-RNAi loser cells might cause elevated lactate production and secretion (Fig. 4a). To detect relative differences in lactate levels in loser and winner cells, we used a

genetically encoded lactate reporter (*UAS-lactate FRET*)[23], which we expressed throughout the entire tissue (*nub-Gal4, UAS-lactate FRET*). Clonal analysis was conducted via the LexA/*lexO* binary system[24]. We found that *NR2*-RNAi loser clones had substantially lower levels of intracellular lactate than surrounding wild-type cells or control clones (Fig. 4b). Importantly, blocking lactate exchange via RNAi-mediated knockdown of *monocarboxylate transporter 1* (*Mct1*) rescued lactate reduction in loser cells. Co-depletion of *Mct1* not only prevented reduction of lactate in loser clones but also caused significant lactate accumulation (Fig. 4c). No such changes in lactate levels were seen under non-competitive control conditions (Fig. 4c). Ex vivo treatment with MCT inhibitors (MCTi) also led to a relative increase of lactate in losers, while it caused a corresponding decrease of lactate in surrounding winners (Fig. 4d). Feeding ᴌ-lactic acid (LLA) to animals, like treating them with MCTi, suppressed lactate reduction in NR2loser clones (Fig. 4e). Together, these data are consistent with the notion that NR2-depleted loser cells are metabolically reprogrammed to produce and secrete lactate.

**The transfer of lactate underpins cell competition**. Next, we tested the role of lactate exchange in loser cell elimination. Because lactate feeding restored intracellular lactate levels in *NR2*-RNAi clones, we tested whether this might block cell competition. Intriguingly, lactate feeding inhibited the elimination of *NR2*-RNAi clones (Fig. 5a–c). Consistent with the notion that lactate transport from *NR2* losers to winners is important for loser/winner relationships and cell competition, we found that blocking loser cells to produce and transport lactate to winner cells rescued loser cell elimination. Accordingly, concomitant down-regulation of the lactate dehydrogenase *ImpL3* in *NR2* loser clones, like co-knockdown of the lactate transporter *Mct1*, rescued the elimination of *NR2* clones (Fig. 5b, c). This effect was specific to MCT1, as silencing the putative monocarboxylate transporters *CG13907* or *CG3409* within loser clones had no effect on *NR2* loser cell elimination (Fig. 5c). Pharmacological inhibition of MCT1 also blocked cell competition and the elimination of *NR2* loser clones (Fig. 5b, c). Consistent with these results, we observed elevated levels of MCT1 in *NR2*-RNAi cells (Supplementary Fig. 7). Together, these data suggest that preventing loser cells from transferring lactate to their neighbours inhibits cell competition.

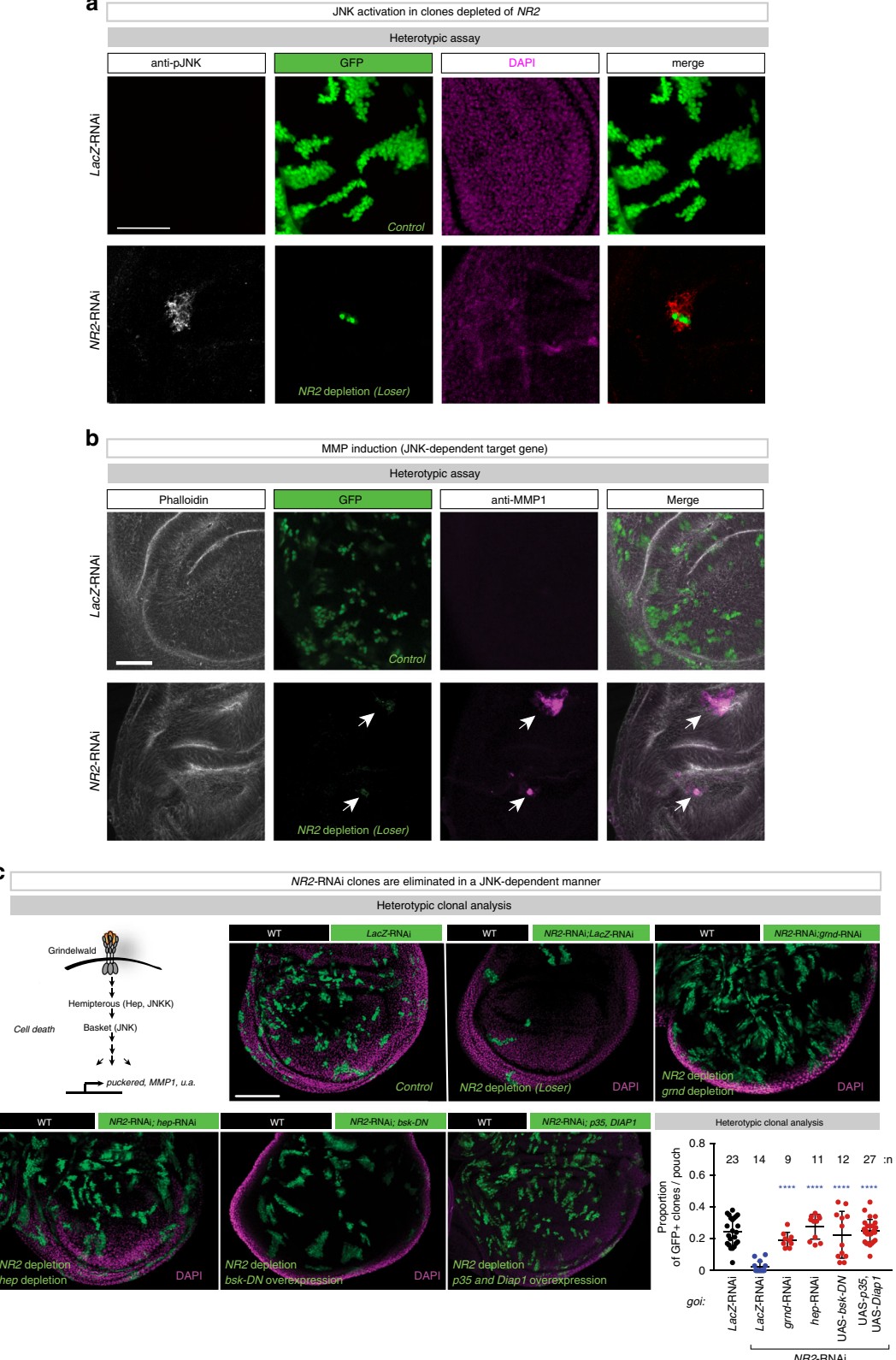

**Fig. 2 NR2 loser clones are eliminated by apoptosis via the TNF>JNK signalling axis. a** Confocal images of wing discs that were immunostained with anti-phospho-JNK. Scale bar 50 μm. **b** Confocal images of wing discs that were immunostained with Phalloidin and anti-MMP1. Scale bar 50 μm. See Supplementary Data Table for genotypes. **c** Heterotypic clonal analysis. Schematic representation of the TNF>JNK signalling pathway. The indicated genes-of-interest (*goi*) were knocked down or misexpressed (*UAS-bsk-DN* or *UAS-p35,UAS-Diap1*) in GFP-marked clones as depicted in Fig. 1b (left panel). Clones are marked by GFP (green). Scale bar 100 μm. Quantification of the heterotypic competition assay. Diagram (error bars) shows the average occupancy of the indicated RNAi/over-expression clones per wing pouch ± SD. ****$P < 0.0001$, ***$P < 0.001$, **$P < 0.01$, *$P < 0.1$ by two-tailed Mann–Whitney nonparametric *U*-test. *n* depicts the number of wing discs. Experiments were repeated four independent times. See Supplementary Table 1 for genotypes.

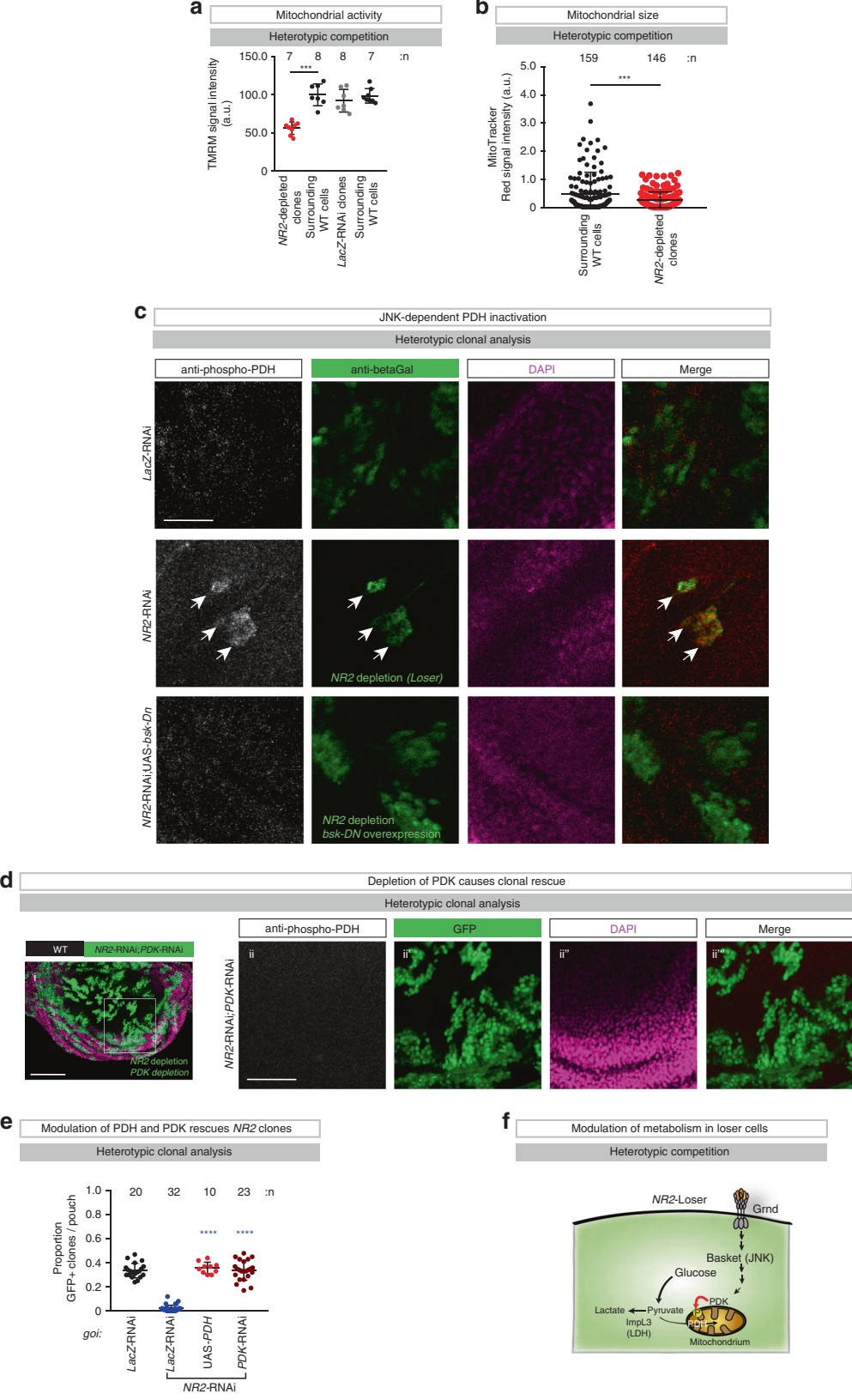

**Lactate-mediated metabolic coupling in Myc-supercompetition**. In mosaic wing imaginal discs, interactions between WT and Myc-expressing cells cause them to acquire 'supercompetitor' behaviour that increases their fitness and enables them to overtake the tissue by killing their WT neighbours[25,26]. To address whether lactate-mediated metabolic coupling might also influence loser/Myc-supercompetitor relationships we first monitored the phosphorylation status of PDH in competing cell populations. Under control conditions, no p-PDH staining was apparent in WT cells (Fig. 6a). However, prominent p-PDH staining was detectable in

**Fig. 3 The TNF>JNK>PDK>PDH signalling axis reprograms the metabolism of NR2 loser clones. a** Quantification of the heterotypic competition assay. Diagram shows the average signal intensity of TMRM staining in RNAi clones and immediately adjacent wild-type cells per wing pouch ± SD. ****$P <$ 0.0001, ***$P <$ 0.001, **$P <$ 0.01, *$P <$ 0.1 by two-tailed Mann–Whitney nonparametric U-test. WT surrounding vs. NR2-RNAi clones ***P value: 0.0003). $n$ depicts the number of cells. See Supplementary Data Table for genotypes. **b** Quantification of the heterotypic competition assay. Diagram shows the average signal intensity of MitoTracker Red staining in RNAi clones and immediately adjacent wild-type cells per wing pouch ± SD. ****$P <$ 0.0001, ***$P <$ 0.001, **$P <$ 0.01, *$P <$ 0.1 by two-tailed Mann–Whitney nonparametric U-test. (NR2-RNAi vs. WT P value: ***0.0007). $n$ depicts the number of cells. See Supplementary Data Table for genotypes. **c** Phospho-PDH-specific immunostaining of clones expressing genes-of-interest (goi) and UAS-LacZ. LacZ expression is revealed by anti-βGal staining. Scale bar 25 μm. Experiments were repeated three independent times. **d** Heterotypic clonal analysis. The indicated gene-of-interest (goi) was knocked down (UAS-PDK-RNAi) in GFP-marked NR2-RNAi clones, and stained for anti-phospho-PDH (ii). Clones are marked by GFP (green). Scale bar 100 μm (i) or 50 μm (ii). **e** Quantification of the heterotypic competition assay. Diagram (error bars) shows the average occupancy of the indicated RNAi clones per wing pouch ± SD. ****$P <$ 0.0001, ***$P <$ 0.001, **$P <$ 0.01, *$P <$ 0.1 by two-tailed Mann–Whitney nonparametric U-test. $n$ depicts the number of wing discs. Experiments were repeated four independent times. See Supplementary Data Table for genotypes. **f** Schematic representation of the metabolic reprograming of loser cells by the TNF>JNK>PDK>PDH signalling pathway.

WT cells that were juxtaposed to Myc winner clones (Fig. 6a–c and Supplementary Fig. 8a, b) but not homotypic controls (Supplementary Fig. 8c). Accordingly, elevated intracellular lactate level was detected in Myc cells juxtaposed to loser clones (Fig. 6d). Consistent with the notion that loser cells produce and transfer lactate to winners, we found that ex vivo treatment with MCT inhibitors (MCTi), which blocks lactate exchange, led to a time-dependent relative increase of lactate in losers, while it caused a corresponding relative decrease of lactate in Myc winners (Fig. 6e). This indicates that Myc supercompetitor cells receive and consume lactate from losers. Accordingly, lactate feeding suppressed Myc-mediated supercompetition (Supplementary Fig. 8d, e). Moreover, we found that pharmacological inhibition of MCT1 suppressed Myc supercompetition (Supplementary Fig. 8d–g). Consistently, we found that Myc clones seemed to have highly active mitochondria whose activity was dependent on NR2 (Fig. 6f). Together, our data indicate that loser cells produce and secrete lactate, while winner clones receive and consume lactate. Our data are in agreement with recent studies[4,27], demonstrating that in vivo some cancer cells preferentially consume lactate instead of generating lactate as a waste product[28].

**NR2 is required for the supercompetitior status of Myc cells.** To test the potential role of NR2 for the supercompetitor status of Myc-expressing cells, we first examined the levels of NR2 in Myc clones. As shown in Fig. 7a–e, Myc supercompetitor clones exhibited higher levels of NR2 than surrounding wild-type neighbours, both in the wing disc as well as fat body. Furthermore, endogenous variation in the expression levels of NR2 (monitored by NR2-Gal4; UAS-GFP) showed a correlation with Myc expression levels (anti-Myc immunostaining) (Fig. 7f–g). The upregulation of NR2 was competition-dependent, as in a homotypic context, no difference in NR2 expression was observed between Myc expressing cells (driven with hh-Gal4) and wild-type cells (Supplementary Fig. 9).

To test whether NR2 contributes to the supercompetitor status of Myc during cell competition, we depleted NR2 in Myc supercompetitor clones. While clonal expression of Myc on its own resulted in large clones, such Myc winners were eliminated when NR2 was simultaneously knocked down in these clones (Fig. 8a, b). Importantly, Myc-expressing NR2 clones were eliminated only in heterotypic settings, when surrounded by WT cells. Accordingly, tissue-wide (nub-Gal4) expression of Myc; NR2-RNAi did not lead to the elimination of GFP + non-competitive clones, demonstrating that such Myc;NR2-RNAi clones are intrinsically viable (Fig. 8c), but when surrounded by WT cells are eliminated via cell competition and caspase-mediated cell death (Fig. 8d). Importantly, under conditions where NR2 was depleted tissue-wide, Myc clones no longer acquired supercompetitor status. Accordingly, wild-type cells

(marked with GFP) were no longer eliminated by Myc clones (Fig. 8e, f). Likewise, treatment with the NMDAR inhibitor AP5 suppressed Myc supercompetition (Fig. 8g, h). These data are consistent with the notion that NR2 contributes to the supercompetitor status of Myc clones.

## Discussion
The elimination of unfit cells via competitive interactions plays an important role for the maintenance of tissue health during development and adulthood[1,2,29–31]. Our data indicate that the NMDA receptor NR2 influences the competitive behaviour of epithelia cells and Myc supercompetitors in the Drosophila wing disc. We find that genetic depletion of NR2 reprograms metabolism via TNF-dependent and JNK-mediated activation of PDK, which in turn phosphorylates and inactivates PDH. This causes a shutdown of pyruvate catalysis and results in a switch to aerobic glycolysis. Upon phospho-dependent inhibition of PDH, pyruvate is reduced to lactate via LDH, and secreted[32]. While lactate exits cells to avoid acidification, it can be recaptured and used as carbon source by other cells, leading to metabolic compartmentalisation between adjacent cells. In normal physiology as well as in murine and human tumours, lactate is an important energy source that fuels mitochondrial metabolism[4,27]. For example, lactate produced and secreted by astrocytes is transported to neighbouring neurons where it is used as source of energy to support neuronal function[33]. This is akin of the 'reverse Warburg effect'[6], also named 'two-compartment metabolic coupling' model, where cancer-associated fibroblasts (CAFs) undergo aerobic glycolysis and production of high energy metabolites, especially lactate, which is then transported to adjacent cancer cells to sustain their anabolic need[6].

Our data suggest that the epithelial NMDA receptor is responsible for fitness surveillance and to provide Myc clones with supercompetitor status. Cells with decreased epithelial NMDA receptor are metabolically reprogrammed to transfer their carbon fuel to their neighbours. According to our model, differential NMDAR signalling in adjacent cells triggers lactate-mediated metabolic coupling, and underpins cell competition in epithelia. Consistently, preventing loser cells from 'transferring' lactate to their neighbours, via inhibition of MCT1, Impl3 or PDK, removes the fitness disparity and nullifies cell competition. Likewise, exposure to elevated levels of systemic lactate, blocks elimination of NR2 loser clones. This suggests that cell competition may be based on NMDAR-mediated metabolic coupling between winners and losers. Importantly, this metabolic coupling only occurs under competitive conditions. Consistently, NR2 losers are only eliminated if they are surrounded by cells with functional NMDAR. This is evident as tissue-wide inhibition of NMDAR by AP5, a selective inhibitor of NR2, blocks elimination of NR2 loser clones in a heterotypic genetic setting.

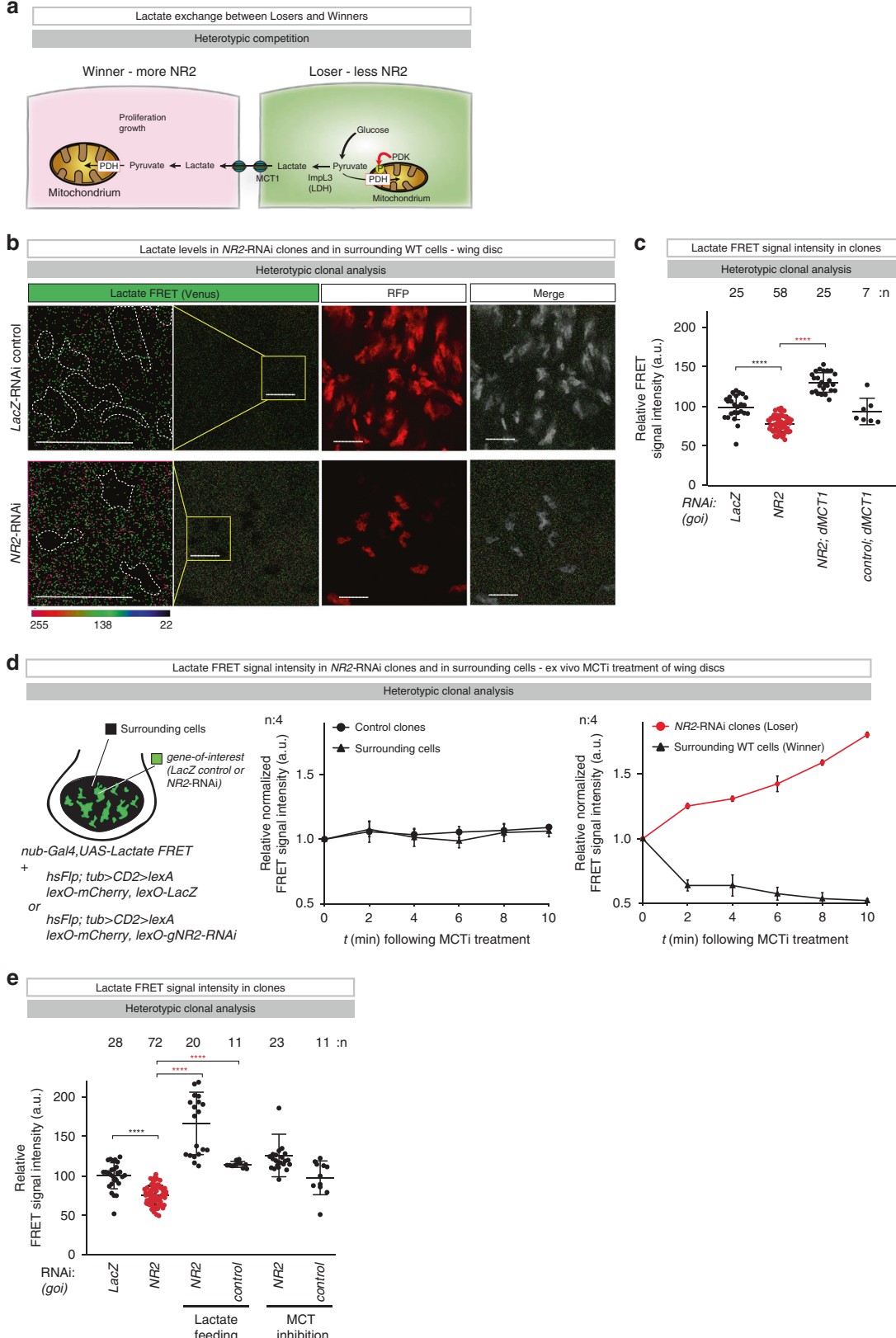

We find that NR2 is upregulated in Myc expressing clones and that Myc cells co-opt epithelial NR2 to promote cell competition, subduing their neighbouring wild-type cells that become re-classified as 'unfit'. Interestingly, Myc clones lose their super-competitor status upon tissue-wide depletion of NR2. Under this condition, WT cells are no longer eliminated and survive among Myc supercompetitors. This indicates that NR2 underpins Myc-induced supercompetition. Given that Myc is a major driver of cancer cell growth, and is a hallmark of the disease in nearly seven out of ten cases, blocking Myc's function would be a powerful

**Fig. 4 Loser cells transfer their lactate to winners. a** Schematic model of NR2-dependent regulation of lactate-mediated metabolic coupling during cell competition. **b** Heterotypic clonal analysis. The indicated genes-of-interest (*goi*) were knocked down. Lactate FRET signal was monitored in clones and in surrounding cells in the wing pouch. Clones are outlined with dashed lines and marked by RFP (red). Scale bars 20 μm. For genotypes see Supplementary Table 1. Experiments were repeated three independent times. **c** Heterotypic clonal analysis. The indicated genes-of-interest (*goi*) were knocked down. Lactate FRET signal was monitored in clones and in surrounding cells in the wing pouch. Diagram shows the relative FRET signal intensity of the indicated RNAi clones per wing pouch ± SD. ****$P < 0.0001$, ***$P < 0.001$, **$P < 0.01$, *$P < 0.1$ by two-tailed Mann–Whitney nonparametric $U$-test. *n* depicts the number of wing discs. For genotypes see Supplementary Table 1. **d** Heterotypic clonal analysis. Relative normalized lactate FRET signal intensity of clonal cells and surrounding cells of the wing pouch was monitored for 10 min following the administration of MCT inhibitor, ex vivo. Diagrams (error bars) show the average of the relative FRET signal intensity of the indicated clones (control or NR2-RNAi) and that of immediately adjacent wild-type cells per wing pouch, as a function of time ± SD. **e** Heterotypic clonal analysis. The indicated genes-of-interest (*goi*) were knocked down. Lactate FRET signal was monitored in clones and in surrounding cells in the wing pouch. Lactate feeding and treatment with MCT inhibitor was conducted as outlined in the Methods section. Diagram (error bars) shows the relative FRET signal intensity of the indicated RNAi clones per wing pouch, using the indicated conditions ± SD. ****$P < 0.0001$, ***$P < 0.001$, **$P < 0.01$, *$P < 0.1$ by two-tailed Mann–Whitney nonparametric $U$-test. *n* depicts the number of wing discs. For genotypes see Supplementary Table 1.

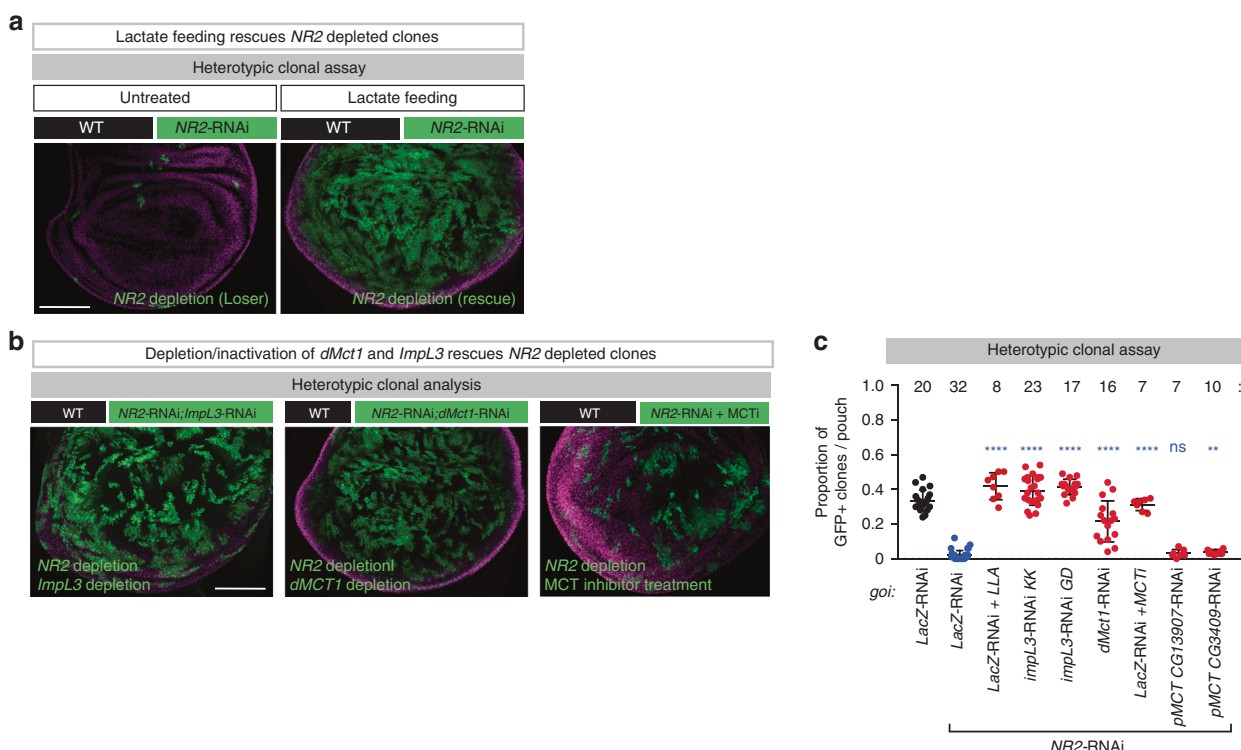

**Fig. 5 NR2 controls cell competition by regulating lactate-mediated metabolic coupling between winners and losers. a, b** Heterotypic clonal analysis. The indicated genes-of-interest (*goi*) were knocked down and over-expressed in *GFP*-marked clones. For genotypes see Supplementary Data Table. Clones are marked with GFP (green). Lactate feeding and treatment with MCT inhibitor was conducted as outlined in the Methods section. Scale bars 100 μm. **c** Quantification of the heterotypic competition assay. Diagram (error bars) shows the average occupancy of the indicated RNAi clones per wing pouch, using the indicated conditions ± SD. ****$P < 0.0001$, ***$P < 0.001$, **$P < 0.01$, *$P < 0.1$ by two-tailed Mann–Whitney nonparametric $U$-test. NR2-RNAi + pMCT CG13907-RNAi vs. NR2-RNAi + LacZ-RNAi P value: 0.0841; NR2-RNAi + pMCT CG3409-RNAi vs. NR2-RNAi + LacZ-RNAi P value: **0.0011). *n* depicts the number of wing discs. Experiments were repeated three independent times. See Supplementary Table 1 for genotypes.

approach to treat many types of cancer. However, the properties of the Myc protein itself make it difficult to design a drug against it. Since the NMDAR signalling circuit is hijacked in many types of human cancers[34], and its expression level is associated with poor patient survival, it is attractive to speculate that targeting NMDAR may be a promising strategy to improve patient care.

## Methods

**Fly strains**. The following strains were used: UAS-LacZ-RNAi[35] (from M. Miura), UAS-dlg1 $^{GD41136}$-RNAi (Vienna *Drosophila* Resource Center, VDRC), NR2:Gfp[36], UAS-scrib$^{TRIP.GL00638}$-RNAi (Bloomington *Drosophila* Stock Centre, BDSC), UAS-NR2$^{TW}$-RNAi[15] and UAS-NR2 (from A.S. Chiang), UAS-NR2$^{TRIPJF02044}$-RNAi (Bloomington *Drosophila* Stock Centre, BDSC), UAS-NR2$^{TRIPHMS02012}$-RNAi

(BDSC), UAS-NR2$^{TRIPHMS02176}$-RNAi (BDSC), UAS-NR2$^{GD3196}$-RNAi (VDRC), UAS-grnd$^{KK109939}$-RNAi (VDRC), UAS-hep$^{GD26929}$-RNAi (VDRC), UAS-bsk-DN (BDSC: 6409), UAS-bsk.B (WT bsk, BDSC: 9310), UAS-p35,UAS-DIAP1 (PMID: 10675329), pucE69(puc-LacZ)[37], Pdk-RNAi$^{TRiPGL00009}$ (BDSC: 35142), UAS-Pdh (BDSC: 58765), UAS-lactateFRET[23] (from B. Hudry), UAS-Mct1$^{KK108618}$-RNAi (VDRC), UAS-impL3$^{GD31192}$-RNAi (VDRC), UAS-impL3$^{KK110190}$-RNAi (VDRC), UAS-CG13907$^{KK107339}$-RNAi (VDRC), UAS-CG3409$^{GD37139}$-RNAi (VDRC), UAS-Myc (BDSC: 9674), nub-Gal4 (BDSC: 25754), hh-Gal4 (BDSC:45946 III), NR2 [MI09281-GFSTF.2] (BDSC: 60566), NR2-Gal4 (BDSC: 76705), nub-LexA (BDSC: 54415 and BDSC: 54963), LexO-NR2-RNAi (generated for this study), LexO-RFP-RNAi (generated for this study), LexO-LacZ (BDSC: 7226), for the generation of GFP-marked clones the following strains were used: y,w,hs-flp;;Act > CD2 > Gal4, UAS-nEGFP[38] (from T. Neufeld). (">" denotes FRT sites) and Ubi-p63E > STOP > Stringer(nEGFP) (BDSC: 32251), w; tub > myc y + >Gal4, UAS-GFP/CyO (from L. Johnston), w; tub>CD2>lexA, lexO-mCherry[39], w; tub > myc > lexA,

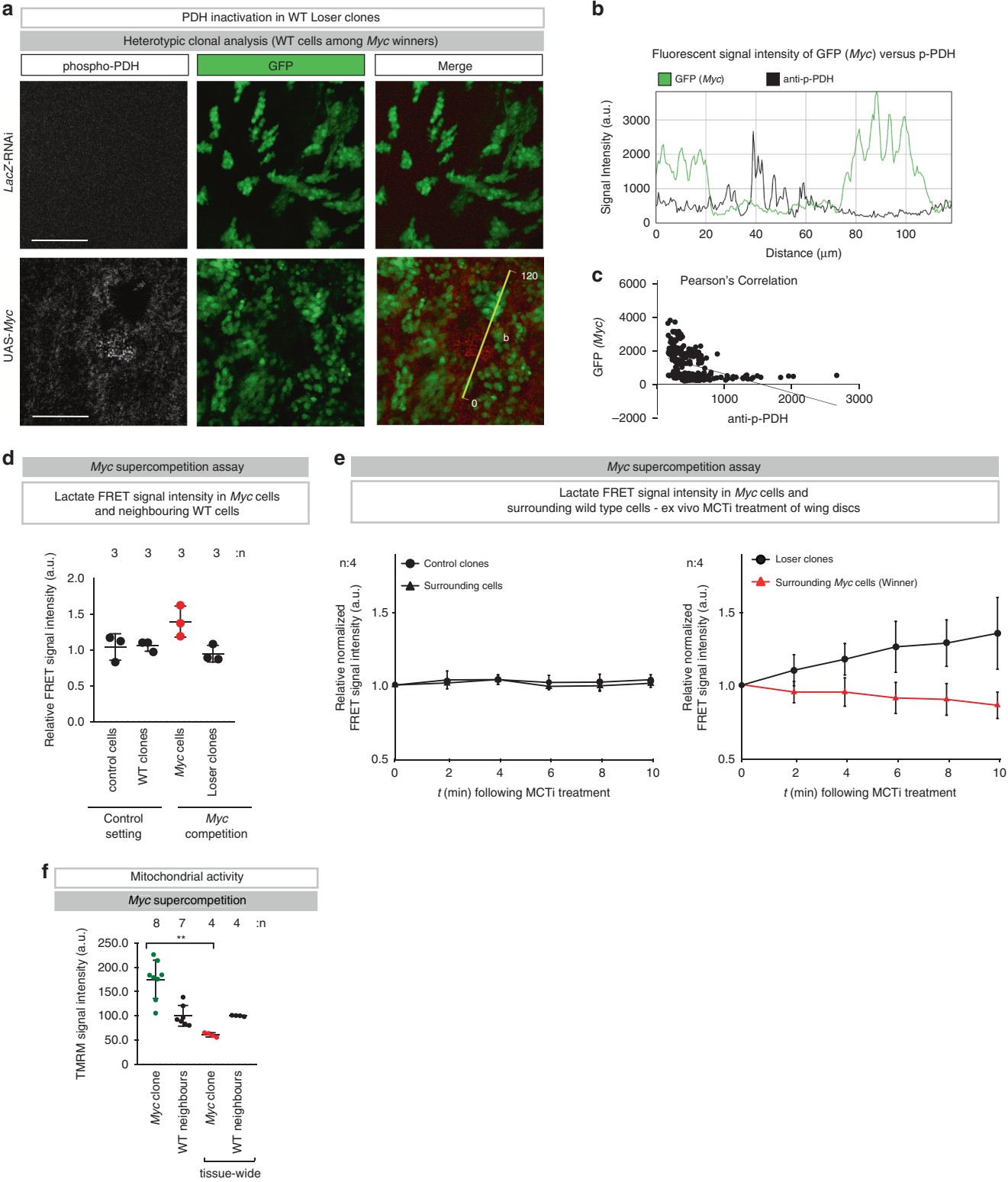

**Fig. 6 Lactate-mediated metabolic coupling occurs during loser and Myc supercompetition. a** Confocal images of dissected wing pouches stained for anti-phospho-PDH (reading out its inactivation). Scale bar 50 μm. Experiments were repeated three independent times. **b** Fluorescent intensities of phospho-PDH (black) and GFP (green) are measured by ImageJ software at the yellow line. **c** Pearson's correlation analysis of phospho-PDH and GFP. **d, e** Heterotypic clonal analysis. Relative normalized lactate FRET signal intensity of clonal cells and surrounding cells of the wing pouch was monitored for 10 min following the administration of MCT inhibitor, ex vivo. See Supplementary Table 1 for genotypes. **f** Quantification of the heterotypic competition assay. Diagram (error bars) shows the average signal intensity of TMRM staining in *UAS-Myc* clones and immediately adjacent wild-type cells per wing pouch ± SD. ****P < 0.0001, ***P < 0.001, **P < 0.01, *P < 0.1 by two-tailed Mann–Whitney nonparametric U-test. (*UAS-Myc* clone with tissue-wide *NR2*-RNAi vs. *UAS-Myc* clone on control P value: 0.004). n depicts the number of wing discs. See Supplementary Table 1 for genotypes.

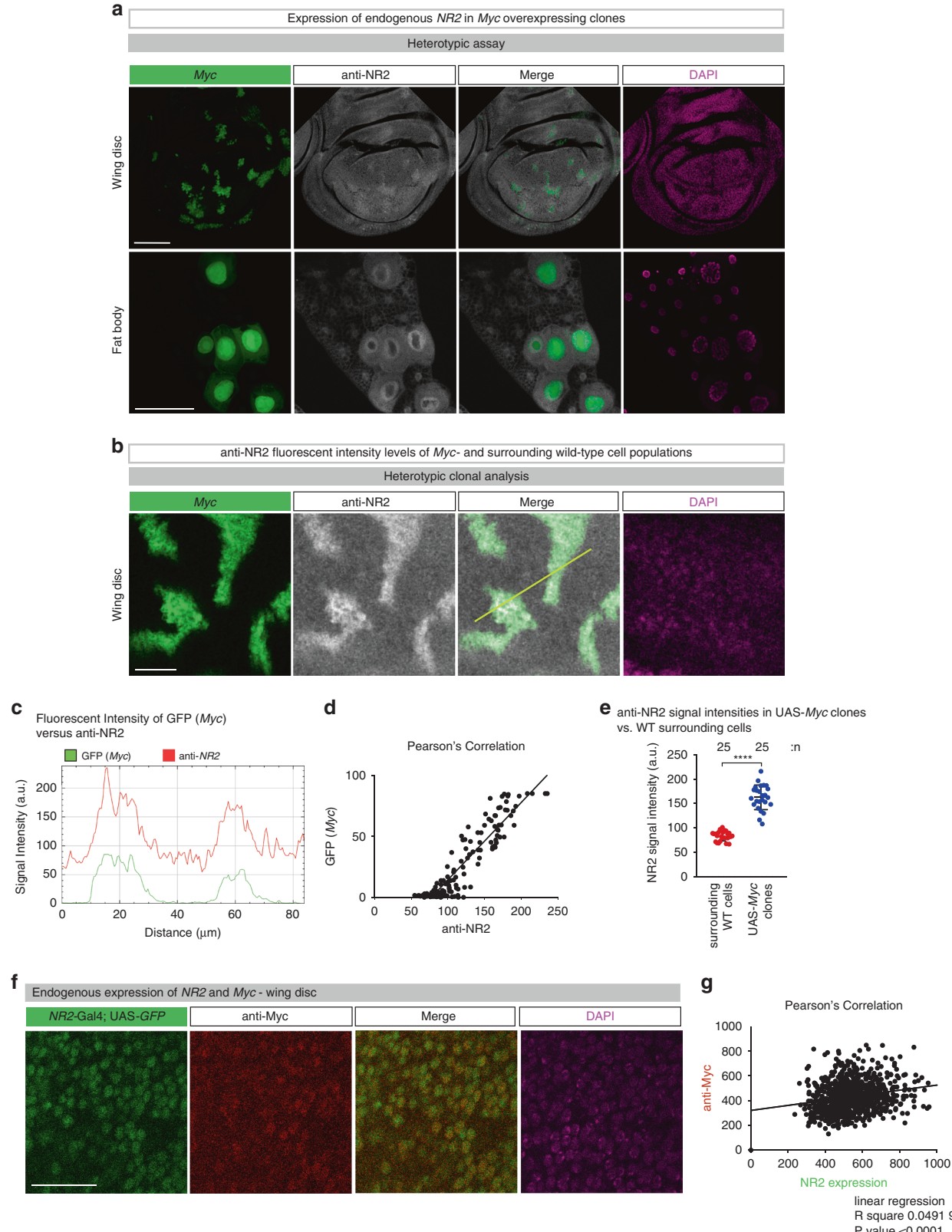

**Fig. 7 NR2 is upregulated in supercompetitior Myc-expressing cells. a, b** Immunostaining of imaginal wing discs and fat body, GFP-marked Myc over-expressing clones. NR2 expression is revealed by antibody specific to the *Drosophila* NR2 subunit. Scale bars 100 and 25 μm, respectively. Experiments were repeated three independent times. **c** Fluorescent intensities of anti-NR2 (red) and GFP (Myc, green) are measured by ImageJ software at the yellow line. **d** Pearson's correlation analysis of anti-NR2 and GFP (Myc). **e** Mean signal intensities (error bars) of anti-NR2 specific staining in Myc expressing cells (blue) and immediately adjacent wild-type cells (red) ± SD. ****$P < 0.0001$, ***$P < 0.001$, **$P < 0.01$, *$P < 0.1$ by two-tailed Mann–Whitney nonparametric *U*-test. **f** Endogenous expression levels of NR2 (followed by GFP) and Myc (anti-Myc immunostaining). Scale bar 20 μm. Experiments were repeated two independent times. **g** Pearson's correlation analysis of NR2 (GFP) and Myc (anti-Myc).

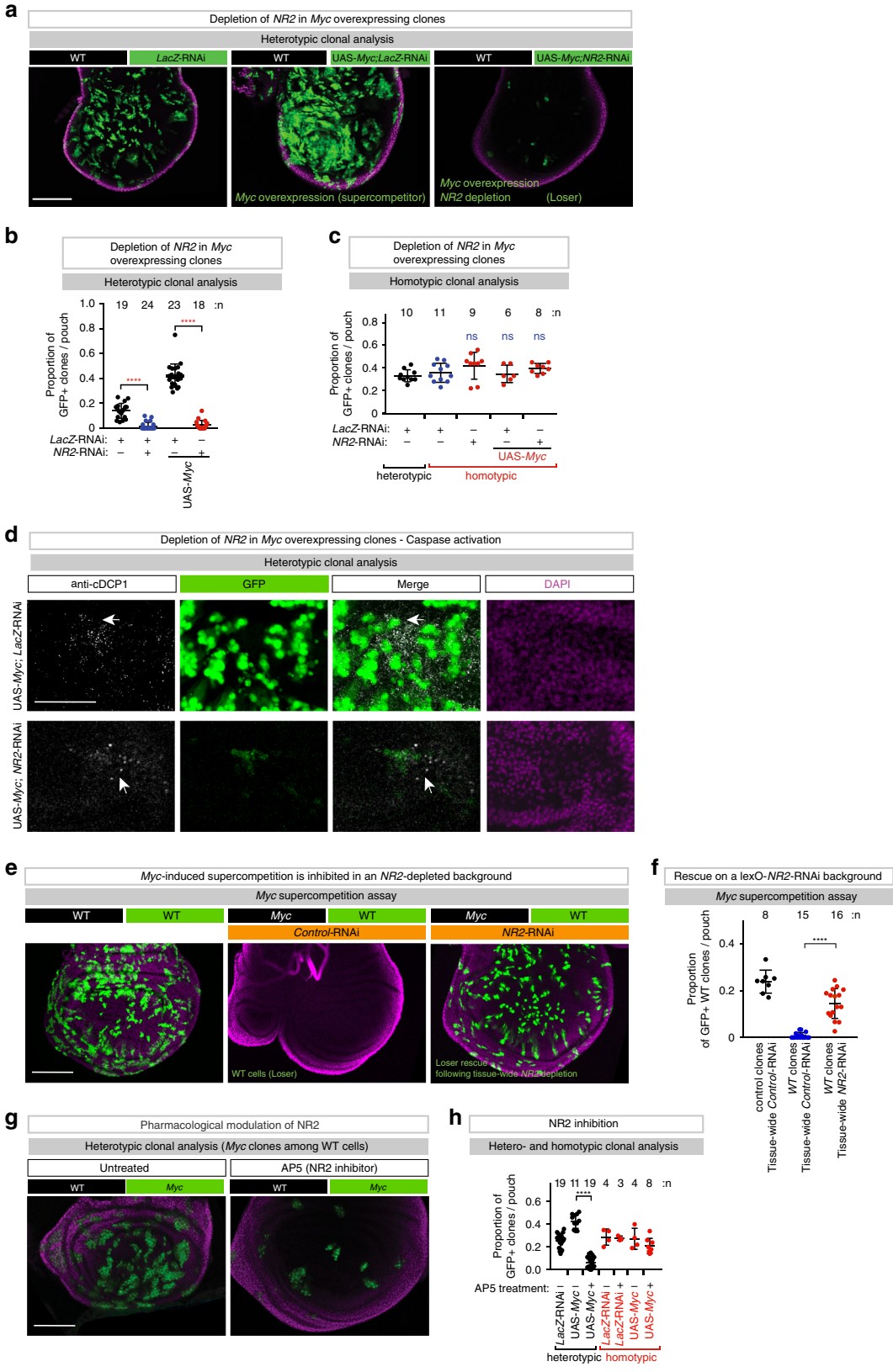

lexO-mCherry[39], and yw; Act > y > UAS-LacZ (from B. Hudry). See Supplementary Table 1 for genotypes.

**Generation of lexO transgenic lines**. In all, 21 bp short hairpin sequences targeting NR2 (CACTTCAAGTTCACTATCCTA) or RFP (CGAGTTCATCTA-CAAGGTGAA) were chemically synthesised and cloned into pJFRC19 vector.

Transgenes were inserted by phiC31-mediated recombination into the attp-9A docking site on the 3rd chromosome (75A10) (BDSC: 9725).

**Fly husbandry**. Fly stocks were reared on a standard corn meal/agar diet (6.65% corn meal, 7.15% dextrose, 5% yeast, 0.66% agar, supplemented with 2.2% nipagin and 3.4 ml/l propionic acid). All experimental flies were kept at RT on a 12 h

**Fig. 8 NR2 is essential for the supercompetitior status of Myc-expressing cells. a** Heterotypic supercompetitor clonal analysis. *Myc*-expressing clones are marked by GFP. *LacZ*-RNAi or *NR2*-RNAi were expressed in clones expressing *Myc*. Experiments were repeated four independent times. **b** Quantification of the heterotypic supercompetitor clonal assays. **c** Homotypic supercompetitor assay. Diagram shows the average occupancy of *Myc* clones on homotypic background. The indicated genes-of-interest were over-expressed (*Myc*) and knocked-down (*LacZ* or *NR2*) throughout the wing pouch. GFP-marked clones represent non-competitive clones. **d** Shown are confocal images of wing discs that were immunostained with anti-cleaved DCP1. Scale bar 50 μm. **e** Heterotypic supercompetitor clonal analysis on *Control*-RNAi or *NR2*-RNAi depleted backgrounds. Wild-type clones are marked by GFP, among Myc expressing cells. Scale bar 100 μm. **f** Quantification of the heterotypic supercompetitor clonal assays. See Supplementary Table 1 for genotypes. **g** Heterotypic clonal analysis in the presence or absence AP5, an inhibitor of NR2. Scale bar 100 μm. **h** Quantification of the AP5 treatment assay. Error bars represent average occupancy of the indicated clones per wing pouch ± SD. ****$P < 0.0001$, ***$P < 0.001$, **$P < 0.01$, *$P < 0.1$ by Mann–Whitney two-tailed nonparametric *U*-test. (b,: *NR2*-RNAi, *UAS-Myc* vs. *NR2*-RNAi,*LacZ*-RNAi *P* value: 0.1247). *n* depicts the number of wing discs. Experiments were repeated three independent times, unless stated otherwise. See Supplementary Table 1 for genotypes.

light/dark cycle. Fly crosses were set up and kept at RT. Flies were transferred to fresh vials every day, and fly density was kept to a maximum of 15 flies per vial. For clonal flip-out experiments (homotypic and heterotypic assays), flies were allowed to lay eggs in fresh tubes for 3 h. In all, 48 h after egg laying (AEL) larvae were incubated at 37 °C for 10 min to induce transgene expression. Following temperature shift, animals were kept at RT. Imaginal wing discs were dissected 48 (for ex vivo lactate FRET experiments) or 72 h after heat-shock-mediated induction of clones.

**Homotypic and heterotypic cell competition assays.** For the homotypic assays, *hsFlp;nub-Gal4;ubi-p63 > STOP > Stringer(nEGFP)* flies were crossed to flies carrying the respective *UAS-based* transgenes. *nub-Gal4* drives expression of the UAS-transgene in the entire wing pouch. Heat shock generates nEGFP-positive non-competitive clones, which have the same genotype as the surrounding wing pouch. For heterotypic assays, *hsFlp;;act>CD2>Gal4,UAS-nEGFP* or *hsFlp; tub>CD2>lexA, lexO-mCherry* flies were crossed to flies carrying the respective *UAS-* or *lexO-based* transgenes. Heat shock generates flip-out clones, removing the *>CD2>* cassette, which allows expression of the *UAS-based* transgenes. Such clones are marked by nEGFP or mCherry expression, respectively. Heterotypic supercompetition assays: based on a similar principle, *hsFlp; tub>myc y+ >Gal4, UAS-GFP/CyO* or *hsFlp; tub>myc >lexA, lexO-mCherry* stocks were crossed to *UAS-* or *LexO-based* transgenes and heat shock generates flip-out clones, removing the *tub>myc y+>* or *>myc>*cassettes, respectively.

**Immunohistochemistry.** Larval tissues were stained using standard immunohistochemical procedures. Briefly, discs were dissected in PBS, fixed at room temperature for 20 min in 3.7% formaldehyde/PBS and washed in 2% Triton-X100/PBS. All subsequent incubations were performed in 2% Triton X-100/PBS at 4 °C. Samples were mounted either in Vectashield or Vectashield containing DAPI (Vector Labs). The following primary antibodies were used: mouse anti-dNR2[15] (1:100, from Ann-Shyn Chiang, mouse anti-dlg1 (1:50, 4F3, DSHB, University of Iowa, Iowa City, IA, USA), mouse anti-GFP (11814460001, Roche), mouse anti-MMP1 (1:50, 5H7B11, DSHB, University of Iowa, Iowa City, IA, USA), rabbit anti-cleaved DCP1 (Asp216) (1:200,Cell Signaling), mouse p-JNK (1:100, 9255,Cell Signaling Technology Inc., Danvers, MA, USA), PHA-555 (Phalloidin-555, A34055, Invitrogen/Molecular probes), mouse polyclonal anti-MCT1 (1:100, ab90582, Abcam), rabbit polyclonal anti-Myc (1:100, d1-717, sc-28207 Santa Cruz Biotechnology), rabbit polyclonal anti-p-PDHE1 (1:200, Pyruvate Dehydrogenase E1-alpha subunit (phospho S293), ab92696, Abcam). Note, the phosphorylation site surrounding S296 of human PDHE1 is conserved in *Drosophila* PDHE1, which is encoded by lethal(1)G0334 (CG7010) (*e*-value 5e-36, query coverage of 99%). Fluorescent secondary antibodies (1:2000, FITC-, Cy3- and Cy5-conjugated) were obtained from Jackson Immunoresearch.

**TMRM and MitoTracker Red stainings.** Mitochondria were stained with 500 nM Tetramethylrhodamine Methyl Ester (TMRM) for 20 min. TMRM staining was recorded with excitation at 543 nm and a 560–615 nm band-pass emission filter. MitoTracker Red CMXRos (M7512, LifeTechnologies) was used at 300 nM in PBS 1% Tween 20 for 10 min.

**Lactate feeding and treatments with inhibitors.** Following heat shock-mediated clone induction, larvae were placed on standard food containing L-lactate (30 mM, 71718-10G, Sigma Aldrich), the NR2 antagonists AP5 (5 μM, A8054, Sigma Aldrich), PEAQX tetrasodium hydrate (0.015 μM, 1999, Sigma Aldrich), Ifenprodil-tartarate salt (15 μM, I2892, Sigma Aldrich) or the MCT1 inhibitor AR-C155858 (100 nM, Bio-Techne (Tocris)) for 48 h. For ex vivo experiments with dissected wing discs, MCT1 inhibitor was added in a final concentration of 10 nM into M16 Medium (M7292, Sigma Aldrich).

**Lactate measurements using FRET-based metabolite sensor.** Imaging experiments were performed using dissected imaginal wing discs. A genetically encoded lactate reporter (*UAS-lactate* FRET)[23], was co-expressed within clones expressing gene-of-interest (*goi*) or *UAS-lacZ-RNAi*. Alternatively, the sensor was expressed

tissue wide, under the control of nubbin (*nub-Gal4 > UAS-lactate FRET*). In the latter case, clonal analysis was conducted via the lexA/*lexO* binary system[24], which allowed side-by-side comparison of the lactate FRET signal in clones expressing the *lexO-goi* or *lexO-LacZ* and surrounding wild-type cells. The dissected tissues were placed into an open μ-slide (chambered coverslip, ibidi #80826). Fluorescent images were acquired via a ×20, ×40 or ×63 objectives on a Zeiss 780 confocal microscope. An Argon laser and MBS458/514 beamline splitter were used with the following filter sets: excitation 458 nm, emission 485-526 nm (mTFP channel); excitation 458 nm, emission 526–625 nm (FRET channel) and excitation 514 nm, emission 526–625 nm (Venus channel). For data analysis, regions-of-interest (ROI) were delimited and the average intensity of both mTFP and Venus channels over each ROI were calculated. As the FRET (from mTFP to Venus) inversely correlates, with lactate concentration, to obtain a signal that positively correlates with lactate concentration, the signal intensity of mTFP was divided by Venus signal intensity.

**Quantifications.** Imaginal discs were imaged at ×20 magnification. Seven Z-stacks were taken for each disc. After imaging, channels were split and maximum Z-projection was analysed. Using DAPI channel images, a line was drawn around the pouch area and measured using ImageJ. The sum areas of GFP-positive clones were measured using the GFP channel. Threshold was adjusted with the Huang auto-thresholding algorithm to subtract background. The area above the threshold was analysed. Data were collected from at least three independent experiments, and 10 wing discs per genotype and/or condition were analysed, unless stated otherwise. The relative occupancy of GFP-positive clones was quantified, and expressed as proportion of GFP-positive clones per pouch (±S.D.).

**Statistics and data presentation.** All statistical analyses were carried out using GraphPad Prism7. Comparisons between two genotypes/conditions were analysed with the Mann–Whitney nonparametric two-tailed rank *U*-test or Pearson's correlation test. Confocal images belonging to the same experiment, and displayed together, were acquired using the same settings. For the purpose of visualization, the same level and channel adjustments were applied using ImageJ. Of note, all quantitative analyses are carried out on unadjusted raw images or maximum projections. Values are presented as average ± standard deviation (S.D.), *P*-values from Mann–Whitney *U*-test (non-significant (ns): $P > 0.05$; *: $0.05 > P > 0.1$; **: $0.1 > P > 0.01$; ***: $0.01 > P > 0.001$;****: $P > 0.0001$ and from Pearson's analysis α = 0.05.

**Reporting summary.** Further information on research design is available in the Nature Research Reporting Summary linked to this article.

## Data availability
The authors declare that all data supporting the findings of this study are available within the article and its Supplementary Information files or from the corresponding author upon reasonable request.

The original data underlying the following figures are provided as a Source Data file: Figs.1d, 1f, 1g, 1i, 2c, 3a, b, 3e, 4c–e, 5c, 6b–f, 7e, 8b, c, f, h and Supplementary Figs. 3, 5, 7c–e, 8d, e, 8f, g.

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

## Acknowledgements
We would like to thank Laura Johnston, Ann-Shyn Chiang, M. Miura and Bruno Hudry for reagents, and Marta Fores Maresma, C. Castilho Soares, Rebecca Wilson, Sidonie Wicky John, Tencho Tenev, Celia Domingues, Katalin Schlett and the iBV PRISM imaging facility for technical assistance. We would like to thank members of the Meier laboratory for discussions and critical reading of the manuscript. We thank Stephane Noselli and his laboratory for his support during the revision period. A.B. was funded by an EMBO Long Term Fellowship (ALTF-48-2014) and an IDEX-Initiative d'excellence Grant. Work in the Meier lab is funded by Breast Cancer Now (CTR-QR14-007) and Biological Sciences Research (BBSRC) (BB/L021684/1). We acknowledge NHS funding to the NIHR Biomedical Research Centre.

## Author contributions
A.B. conceived the study, A.B. and P.M designed the experiments, and A.B. performed the experiments. A.B. and P.M. analysed the data and wrote the paper.

## Competing interests
The authors declare no competing interests.
