## [Peer Review File · Nature Communications]

Reviewers' Comments:

Reviewer #1:

Remarks to the Author:

Banreti and Meier report a very interesting study of the role of the NMDA receptor in maintaining epithelial cell fitness in *Drosophila*. The larval imaginal discs, epithelial tissues that are the primordia for the *Drosophila* adult appendages, appear to surveil cell fitness throughout their growth, so that cells recognized as relatively unfit are eliminated in a process known as cell competition. Cells mutant for the epithelial polarity genes *dlg* or *scrib*, for example, are eliminated from imaginal discs via cell competition. *Dlg* is a homolog of mammalian PSD-95 and SAP97, which physically interact with the NMDAR subunit NRB2. Although most well known for their role in neural activities in the mammalian brain, NMDA receptors are also expressed in other tissues, with largely unknown functions. The *Drosophila* NMDA receptor consists of two subunits, NR1 and NR2. Banreti and Meier documented expression of NR2 in the larval brain, and found that it is also widely expressed in several non-neural tissues, including the wing disc. The authors thus examined whether NR2 plays a role in cell competition in the wing disc.

The authors used RNAi against NR2 to show that NR2-depleted cells are viable in a homotypic environment but inviable in a mosaic (heterotypic) environment. This is a hallmark feature of cell competition, thus the authors conclude that the death of the NR2-depleted cells is due to cell competition induced upon recognition of disparities in NR2 levels between neighboring cells in the tissue. Through a series of careful and rigorous experiments the authors demonstrate that JNK signaling is activated via the TNFR/Grnd in the NR2-depleted cells, and causes their death via apoptosis. Moreover, they observed a JNK signaling-dependent increase in phosphorylation of pyruvate dehydrogenase (PDH) in the NR2-depleted cells. In its phosphorylated form, PDH is inactive, which leads pyruvate to be converted to lactate, increasing the cell's use of glycolysis to derive energy and reducing mitochondrial activity. Depletion of PDK (which phosphorylates PDH), or expression of PDH in the NR2-depleted clones prevented their death and eliminated the p-PDH staining, suggesting that PDH inactivation of by phosphorylation mediated the death of the NR2-depleted cells. Since p-PDH would increase production of lactate in the NR2-depleted cells, allowing it to be secreted and taken up by nearby wildtype cells, they reasoned that this could potentially fuel the competition. To test this idea, the authors normalized lactate levels across the tissue by feeding the larvae L-lactic acid (LLA) and carried out other experiments to block lactate exchange. All of the treatments prevented the elimination of NR2-depleted cells, suggesting that lactate production and secretion are critical events in the elimination of NR2-depleted cells in the mosaic discs. The authors conclude that clonal (mosaic) NR2 loss activates JNK signaling, leading to metabolic changes that results in increased lactate production and secretion, and ultimately to the death of the cells.

The authors also tested whether NR2 plays a role in Myc-induced super competition, a well characterized form of cell competition. They found that NR2 expression is increased in Myc expressing cells (it isn't clear if this is only when Myc is expressed in clones or also when Myc is expressed homotypically).

[Redacted]

Overall the experiments have been carefully done and the data presented here are very interesting and compelling in terms of the effects of NR2 depletion, which are clearly an outcome of the heterotypic background since it does not occur when NR2 is depleted from all cell, thus it undoubtedly represents a form of cell competition. However, the logic seems to fall apart in going from the data to their proposed model, especially when it involves Myc *[Redacted]*. (Indeed, the model proposed on page 10 and in Fig. 4A is confusing, as it seems to represent generic cell

competition, although in the legend it is stated to represent NR2-depletion induced cell competition. Shouldn't this be more explicitly shown in the image itself?) A more parsimonious view of their data is that loss of NR2 in clones of Myc- [Redacted] expressing cells results in synthetic lethality, perhaps in part from stresses invoked in a heterotypic environment.

The discussion section brings up many interesting ideas about lactate exchange between cells and metabolic coupling and how this happens in other cell types and could work in tumors, but whether this occurs in contexts of cell competition other than in NR2 depletion in mosaic tissues is not clear. The authors state: "Our data suggest that epithelial NMDA receptor activity is responsible for fitness surveillance and to provide clones with oncogenic mutations supercompetitor status." Based on their model they seem to be extrapolating that during Myc super competition, differences in NR2 expression between wildtype cells and Myc super competitor cells causes similar effects, and that the increase in lactate secretion by the wildtype cells (with less NR2) fuels the growth of the Myc super-competitor cells. Are the authors proposing that the differences in NR2 expression between Myc expressing cells and wildtype cells underlies the "fitness" difference between the cells? And promotes lactate-mediated metabolic coupling between the cells and cell competition? Myc-expressing cells have been shown to make more lactate as part of an increase in aerobic glycolysis due to the Myc-induced transcriptional program that is enhanced during super competition (de la Cova et al, 2014). Are they proposing that these cells take up more lactate from their neighboring wildtype "loser" cells? Although the data on NR2 seems very solid and interesting, the tendency to try to generalize the ideas is confusing and needs to be addressed.

Specific comments:

1. At the end of p6, the authors state that NR2-depleted cells are "actively eliminated" yet this is not shown until the next section, where they show data that indicates that Grnd JNK signaling leads to cell death. (e.g., it could have been cell cycle arrest, which is not 'active' elimination).

2. [Redacted]

3. Fig. S6. It is very difficult to see any non-GFP-positive cells in the panels in B. Although they show a trace of the fluorescence (presumably of this disc) in C, the images in B are difficult to reconcile with it. An image of a different disc with fewer clones would better serve the point that the authors are trying to make here. The image of the DAPI channel is also difficult to see/interpret.

4. Fig. S7. The authors state on page 9 that clones lacking NR2 and expressing Myc [Redacted] are eliminated, yet they also state that such clones express p-PDH (a large such clone is shown in Figure S7). This is confusing, would the authors please explain this discrepancy? [Redacted]

5. Is NR2 expression increased in all Myc expressing cells – or is it dependent on clonal expression of Myc? Is glutamate also increased in wing disc cells, or just in FB?

6. On p. 8 it is stated, "...preventing loser cells from transferring lactate to their neighbours may remove fitness disparities and inhibit cell competition." Do wildtype cells take up lactate from each other during normal growth?

7. The increase in p-PDH in the NR2-depleted clones led to a decrease in mitochondrial size as assessed by Mitotracker red staining. Is a decrease in mitochondrial size important? Do mitochondria shrink with decreased activity?

8. Does activation of TNFR/JNK in clones of cells lead to an increase in p-PDH? The implication

from their model is that it would. Or does this metabolic reprogramming only occur when NR2 is depleted?

9. In Fig. 6A, B, the images of the DAPI channels here are oddly restricted to the periphery of the discs, making it difficult to look at the integrity of the disc. Could the authors provide better images of the DAPI channel? Also, the disc with Myc expressing clones in A shows a puzzling whirling morphology of the clones that is very unusual for Myc expressing clones. Is this representative of all their discs with Myc clone?

10. Fig. 6F. The control (lacZ-RNAi) should be in Myc-expressing clones, not just GFP-expressing clones, in order to compare with the Myc-expressing, NR2-expressing clones.

Minor/grammatical

1. The authors begin the paper by saying they carried out a candidate based approach to finding regulators of cell competition, however no other "candidates" are tested or mentioned. On the other hand, they present quite a detailed description of NMDA receptors and interactors by way of a rationale for studying NR2 - thus they don't need to bring in the candidate approach as justification for their study.

2. A reference for NR2:GFP is needed (Nagarkar-Jaiswal et al., 2015, eLife 4: e05338).

3. On p. 6, why is the term "fitness disparity" in quotes and italics?

4. On p. 8, a typo: PHD (instead of PDH)

5. Fig. 3.

Panel A is a little misleading - the loser clone here is expressing NR2-RNAi, which leads to JNK activation in those cells, which according to their results is required to alter the cells' metabolism. Or, as mentioned above, is Grnd and JNK activation sufficient to cause the metabolic programming?

6. I have issues with the authors' use of the term "loser", throughout the manuscript and Figures. For example, the authors state: "Importantly, RNAi-mediated depletion of NR2 or Dlg1 had no effect on clonal survival under homotypic condition (losers among losers), such as upon tissue-wide NR2 depletion using nubbin-Gal4 or hedgehog (hh)-Gal4". While realizing that their descriptors (losers among losers and losers among winners) is an attempt to clarify, it actually makes things more confusing - It is important to remember that cells are only "losers" when in a heterotypic environment. In a homotypic environment, the cells are not "losers" and shouldn't be called that.

7. Several figure legends say "Loser clones are marked by GFP", however, the control clones are also marked by GFP, and these are not "losers". Some of the legends also say: "The indicated genes-of-interest (goi) were knocked down and over-expressed in GFP-marked clones". This is more accurate - they don't need to say 'Loser' clones.

Reviewer #2:

Remarks to the Author:

Banretti and Meier Characterize the role of the NMDA receptor in cell competition. They show that signaling through this receptor mediates cell competition and that glucose/lactate metabolism and lactate transport plays a crucial role in NR2-mediated cell competition. Finally, they show that NR2 elimination in clones of Myc [Redacted] overexpressing cells reverts the winner phenotype of such clones. Therefore, this work presents three different stories, one showing the role of NMDA receptors in cell competition, one showing the role of lactate transport in cell competition and a third investigating the role of NMDA receptors in Myc [Redacted] induced supercompetition. While the three stories are new and exciting, each of them has not been developed to a sufficient level of understanding, which is somewhat frustrating. Surprisingly, the title alludes to the relationship between Myc and NMDA which, in my opinion, is the less developed story. At least one of these

stories needs to be developed further to make a compelling story and the title and abstract should be tuned accordingly.

1-Regarding the NMDA story, there is no insight on how the activity of the channel (across which, cations can enter or exit the cell) impacts on cell fitness. In addition, it is not clear how the higher production of glutamate (intracellular, as judged by the images) impacts the pathway. Furthermore, it is not clear whether during cell competition winner cells activate NR2 or loser cells downregulate it and whether these expression changes are cell-autonomous or depending on the heterotypic context.

2-Regarding the lactate transport hypothesis, while the authors clearly demonstrate the involvement of lactate production and transport for loser status acquisition, the proposed model does not explain the results. The authors propose that donation of Lactate from losers renders winner cells fitter because they obtain extra fuel. This idea is difficult to accept, provided that lactate transport is never against gradient and therefore lactate should always be more concentrated in loser cells (although this has not been directly measured), which, under the author's hypothesis would render them fitter. Furthermore, what triggers loser cell death under this hypothesis remains unknown. If it was JNK activation, then the whole Lactate story would not hold, as in principle it would work downstream JNK activation. Previous involvement of Warburg metabolism and Lactate production has been described in MDCK cells, so further elaboration would be needed here.

3-Regarding the effect of PDH/PDK, if PHD regulation is at the post-translational level, why does PHD mRNA overexpression completely rescue loser clones and how is it that p-PDH is not detected in these clones?

4-Regarding the Myc [Redacted] connection, the authors demonstrate dominance of the loser phenotype induced by NR2 elimination over Myc [Redacted]. This is an interesting concept, especially in understanding and intervening tumor growth, however the NMDA pathway has already been described in this context. The observations reported here show that NMDA heterotypic loss leads to death despite Myc [Redacted] but this does not necessarily mean that differential NR2 activity mediates Myc [Redacted] cell competition, just dominance of one phenomenon versus the other. Functional demonstration should involve showing that during Myc [Redacted] induced competition loser cells are eliminated due to differential NR2 activity. The easiest way to do this would be to induce Myc [Redacted] clones in a homotypic NR2-deficient background or in the presence of NMDA inhibitors. Such experiments would show whether inability to produce imbalance in NR2 activation between winner and loser cells blocks Myc [Redacted] induced competition. In the absence of these characterizations, the conclusions presented and the title do not clearly connect to the presented evidence. Actually, several statements in the abstract do not correspond with any experiment presented in the manuscript, like "Pharmacological inhibition or genetic depletion of NR2 changes the supercompetitor status of oncogenic Myc [Redacted] clones into 'superlosers', resulting in their elimination via cell competition by wild-type neighbours in a TNF-dependent manner." I have not seen in this manuscript any results involving TNF or NR2 pharmacological inhibition in Myc-mosaic studies.

5 [Redacted]

Minor point: The intro on NMDA receptors in the first part of the results could be moved to intro

Reviewer #3:

Remarks to the Author:

Banreti and Meier present a very interesting and potentially significant report describing how the differential expression/activity of NMDA receptors in different cell populations in drosophila can decrease the survival of the weakly activated cell by promoting lactate shuttling to the competing cell. They show that the oncogenes myc [Redacted] appear to use this mechanism to out-compete normal cells.

Specific comments:

- 1) For the non-specialist, just how is the "Proportion of GFP+ clones / pouch" determined? Are these cells or cluster of cells? If cluster of cells, how are adjacent clusters distinguished from a larger cluster and how are the total number of clusters determined? If single cells, are these expressed as the fraction of DAPI+?
- 2) Was the Proportion of GFP+ clones / pouch analysis performed blind to the conditions?
- 3) Why did the authors pick the concentrations for the MCT-1 and NMDA receptor inhibitors that they did? The MCT-1 inhibitor is ~ 100 fold more potent at its target than the NMDA receptor inhibitor for its target, yet the MCT-1 inhibitor was used at a 20-fold higher concentration. Given that these inhibitors were provided in the food, the final target concentration would be very difficult to predict. Is there other literature or experimental evidence that these concentrations are sufficient?
- 4) Do the authors have any speculation as to how the differential activation of NMDA receptors in the two cell populations leads to JNK signaling?
- 5) p. 5, the role of NMDA receptors in regulating lactate neuronal/astrocyte coupling is currently not widely accepted whereas the other actions are well established.
- 6) top of p. 8. The authors note that overexpressing PDH in NR2 loser clones prevented their elimination and prevented phospho-PDH staining. I follow why excess PDH may protect the cells, but why would phospho-PDH staining go down? If the loser clones with or without extra PDH had similar levels of active PDK, then one might expect the PDH/phospho-PDH ratio to change but the phospho-PDH levels might be the same (or higher, given more substrate)?
- 7) typos: P9, line 2, suppercompetitor; P9, line 5, need not capitalize "Glutamate"; P10, line 4, "may influences".
- 8) In the summary, "Myc cells upregulate NMDAR2 (NR2) to gain...". It might help some readers to specify that myc cells upregulate their NMDAR2 levels... since there is a possibility that myc cells somehow alter NMDAR2 levels in non-mutant cells.

Response to Reviewer comments on ms NCOMMS-18-14726-T

We would like to thank you and all the referees for the helpful comments and constructive criticisms. We have responded to them in full with additional experiments and a redesign of the ms that conveys a simple message. We also have corrected all technical shortcomings.

→ Key points raised by the reviewers:

While the reviewers found our story 'new and exciting', reviewer 1 and 2 expressed their concern regarding our proposed 'lactate transfer model' (reverse Warburg effect). In particular, they questioned the possibility that winner and cancerous supercompetitor cells (Myc cells) would use lactate as energy source. They based their argument on the Warburg effect of cancer cells, which dictates that cancer cells have defective mitochondria and hence favour metabolism via aerobic glycolysis rather than the much more efficient oxidative phosphorylation pathway, which is the preference of normal cells of the body (see Illustration below). According to Warburg, cancer cells would produce high levels of lactate, which is then secreted. Accordingly, in the past lactate has been regarded more as a metabolic waste product than a fuel in tumour cells.

Change of Dogma. Recent studies in *Cell* and *Nature* described the metabolism of lactate in human lung tumours *in vivo* and experimental tumours in mice, showing that in certain tumours exogenous lactate is consumed and used, predominantly over glucose, as a respiratory fuel¹⁻³. Moreover, increasing evidence demonstrate that the '**reverse Warburg**' effect, also referred as 'two-compartment metabolic coupling' model, is common in cancers⁴⁻⁶. Several studies have proposed that tumours use lactate as a fuel, expanding its metabolic functions in cancer. Lactate circulates at levels of 1–2 mM and acts as an interorgan carbon shuttle in mammals⁷. Some cancer cells use lactate as a respiratory substrate and a lipogenic precursor in culture⁸. Blocking lactate uptake with an MCT1 inhibitor reduces respiration and promotes glycolysis in some cancer cell lines and suppresses xenograft growth in mice^{9,10}. In mouse models of breast cancer, the growth-promoting effect of stromal cells is impaired by glycolytic inhibition, suggesting that the stroma provides nutritional support to malignant cells by secreting lactate^{9,11,12}. These observations demonstrate that **lactate can serve as fuel** to complement glucose metabolism in tumours.

→ Our new observations are entirely consistent with these findings:

- 1) First, we have consolidated our observation that relative levels of NR2 determines cell competition:
 - b. Tissue-wide depletion of NR2 has no effect on cell viability and growth (*Fig. 1e,f,g*).
 - c. Clonal depletion of NR2 results in their rapid elimination via the TNFR/Grnd>JNK signalling pathway (*Fig. 1c,d,g* and *Fig. 2*).
 - d. Local over-expression of NR2 causes NR2 cells to acquire supercompetitor-like behaviour that enables them to overtake the tissue (**new data Fig. 1h,i**).
- 2) We find that the JNK>PDK signalling axis in NR2-depleted 'loser' cells results in phosphorylation and inactivation of PDH (*Fig. 3c,d,e,f*), the enzyme that converts pyruvate to Acetyl-CoA, which is then used in the TCA cycle of mitochondria. Accordingly, NR2-depleted loser cells have less active and smaller mitochondria (**new data Fig. 3a,b,f**), leading to metabolic reprogramming of such loser cells.
- 3) Using a FRET-based lactate sensor to measure lactate in winners and NR2-losers we found that NR2-depleted loser cells produce lactate, which is then passed on to adjacent winners to render them supercompetitor. Blocking lactate uptake with an MCT1 inhibitor reduces lactate levels in winners over time, and concomitantly results in lactate accumulation in losers (**new data Fig. 4a,b,c,d,e**). Therefore, our data are entirely in line with the current literature on the reverse Warburg effect.
- 4) Preventing lactate transfer from losers to winners, via lactic acid feeding, clonal depletion of *MCT1* or *Impl3* (*Fig. 5a,b,c*), abrogates NR2-mediated cell competition.
- 5) We find that lactate-mediated metabolic coupling also influences loser/Myc-supercompetitor relationships (**new data Fig. 6a,b,c,d,e,f**).
- 6) NR2 is hijacked in Myc-based supercompetition. Myc cells upregulate NR2 (*Fig. 7a,b*) and critically depend on it for their supercompetitor status (*Fig. 8a,b,c,d*). Importantly, in a tissue that globally lacks NR2, Myc clones fail to acquire super-competitor status (**new data Fig.**

8e,f,g,h).

- Together our data are consistent with the notion that NMDAR underpins cell competition and that targeting NMDAR prevents oncogenic Myc clones to acquire supercompetitor status.
- As our revised ms has been changed considerably, we have abstained from highlighting the respective text changes as they would have been too considerable. We refer the reviewers to the revised manuscript.
- Please find below a point-by-point response to the reviewers' comments, with the reviewers comments in blue boxes and our response in 'plain text'.

Reviewer 1:

We would like to thank Reviewer 1 for all the constructive criticism, which certainly have improved our ms. The key aspect to address was the involvement of the reverse Warburg effect in cell competition. As outlined further below, and in our fully revised ms, our new data demonstrate that loser cells produce and secrete lactate, while winners take up and consume lactate.

- It is important to point out that our data do not contradict de la Cova et al. (2014) because enhanced glucose uptake could also occur. Many tumour cells can consume both glucose as well as lactate, and can readily switch between the two carbon sources.

Overall the experiments have been carefully done and the data presented here are very interesting and compelling in terms of the effects of NR2 depletion, which are clearly an outcome of the heterotypic background since it does not occur when NR2 is depleted from all cell, thus it undoubtedly represents a form of cell competition.

To further corroborate our data, we examined the consequence of clonal over-expression of NR2. As shown in **new Fig. 1h,i**, NR2 over-expressing clones overgrow at the expense of wild-type surrounding cells. This suggests that forced expression of NR2 causes NR2 cells to acquire supercompetitor behaviour that enables them to overtake the tissue. Together, our data indicate that relative levels of NR2 underpins cell competitive behaviour in the wing epithelia.

New Fig. 1h,i

However, the logic seems to fall apart in going from the data to their proposed model, especially when it involves Myc [Redacted]. (Indeed, the model proposed on page 10 and in Fig. 4A is confusing, as it seems to represent generic cell competition, although in the legend it is stated to represent NR2-depletion induced cell competition. Shouldn't this be more explicitly shown in the image itself?)

We have amended the model, and now clearly indicate that it refers to NR2-based cell competition (**new Fig. 4a**).

New Fig. 4a. Schematic diagram depicting the reverse Warburg effect during winner/loser relationships in NR2-mediated cell competition.

- [Redacted]

A more parsimonious view of their data is that loss of NR2 in clones of *Myc- [Redacted]* expressing cells results in synthetic lethality, perhaps in part from stresses invoked in a heterotypic environment.

To address this issue we knocked down NR2 tissue-wide and generated *Myc* clones. Importantly, in a tissue that globally lacks NR2, *Myc* clones (unmarked) are not eliminated but lose supercompetitor status (new Fig. 8e,f). Accordingly, wild-type cells (marked with GFP) were no longer eliminated by *Myc* clones. This demonstrates that NR2 contributes to the supercompetitor status of *Myc* clones.

New Fig. 8

Moreover, we have also tested the effect of AP5 (an NMDAR inhibitor) on *Myc*-mediated cell competition. Interestingly, we find that pharmacological inhibition of the activity of NMDAR suppresses the supercompetitor status of *Myc* (new Fig. 8g,h). These data are consistent with the notion that the activity of NMDAR mediates cell competition.

New Fig. 8g,h

The discussion section brings up many interesting ideas about lactate exchange between cells and metabolic coupling and how this happens in other cell types and could work in tumors, but whether this occurs in contexts of cell competition other than in NR2 depletion in mosaic tissues is not clear.

The authors state: "Our data suggest that epithelial NMDA receptor activity is responsible for fitness surveillance and to provide clones with oncogenic mutations supercompetitor status." Based on their model they seem to be extrapolating that during *Myc* super competition, differences in NR2 expression between wildtype cells and *Myc* super competitor cells causes similar effects, and that the increase in lactate secretion by the wildtype cells (with less NR2) fuels the growth of the *Myc* super-competitor cells. Are the authors proposing that the differences in NR2 expression between *Myc* expressing cells and wildtype cells underlies the "fitness" difference between the cells? And promotes lactate-mediated metabolic coupling between the cells and cell competition? *Myc*-expressing cells have been shown to make more lactate as part of an increase in aerobic glycolysis due to the *Myc*-induced transcriptional program that is enhanced during super competition (de la Cova et al, 2014).

Are they proposing that these cells take up more lactate from their neighboring wildtype "loser" cells? Although the data on NR2 seems very solid and interesting, the tendency to try to generalize the ideas is confusing and needs to be addressed.

Our data do not contradict the work of de la Cova et al because many tumour cells can consume both glucose as well as lactate, and can readily switch between the two carbon sources. However, our new data using a FRET-based lactate sensor clearly demonstrates that winner cells consume lactate that is passed on from loser cells. Accordingly, *ex vivo* treatment with MCT inhibitors (MCTi) leads to a relative increase of lactate in losers, while it causes a corresponding decrease of lactate in surrounding winners (Fig. 4d). Together, these data are consistent with the notion that NR2-depleted loser cells are metabolically reprogrammed to produce and secrete lactate. See our **new data Fig. 4a,b,c,d,e**

Importantly, interference with this lactate transfer completely blocks cell competitive behaviour. See previous Fig. 5a,b,c.

Fig. 4d (Right): NR2-loser cells transfer their lactate to winners. Lactate FRET signal was monitored in clones and in surrounding cells in the wing pouch. Wing discs were cultured *ex vivo* and Lactate FRET signal measured over time following treatment with an MCT inhibitor.

Likewise, lactate-mediated metabolic coupling also seems to influence loser/Myc-supercompetitor relationships. Elevated intracellular lactate level was detected in Myc cells juxtaposed to loser cells (**new Fig. 6d**). Consistent with the notion that loser cells produce and transfer lactate to winners, we found that *ex vivo* treatment with an MCT inhibitor (MCTi), which blocks lactate exchange, led to a time-dependent increase of lactate in losers, while it caused a corresponding decrease of lactate in Myc winners (**new Fig. 6e**). This indicates that Myc supercompetitor cells receive and consume lactate from WT losers.

New Fig. 6d. Lactate-mediated metabolic coupling influences loser/Myc-supercompetitor relationships. Lactate FRET signal was monitored in clones and in surrounding cells in the wing pouch. Wing discs were cultured *ex vivo* and Lactate FRET signal measured over time following treatment with an MCT inhibitor.

Specific comments:

1. At the end of p6, the authors state that NR2-depleted cells are “actively eliminated” yet this is not shown until the next section, where they show data that indicates that Grnd JNK signaling leads to cell death. (e.g., it could have been cell cycle arrest, which is not ‘active’ elimination).

This has been corrected. We now indicate: “...resulting in the loss of otherwise viable cells”.

2. [Redacted]

3. Fig. S6. It is very difficult to see any non-GFP-positive cells in the panels in B. Although they show a trace of the fluorescence (presumably of this disc) in C, the images in B are difficult to reconcile with it. An image of a different disc with fewer clones would better serve the point that the authors are trying to make here. The image of the DAPI channel is also difficult to see/interpret.

This has been corrected. We mistakenly included a 7 stack Z-stack image (previous Fig. S6). It was replaced with a single stack image, see Figure 6a.

4. Fig. S7. The authors state on page 9 that clones lacking NR2 and expressing Myc [Redacted] are eliminated, yet they also state that such clones express p-PDH (a large such clone is shown in Figure S7). This is confusing, would the authors please explain this discrepancy? [Redacted]

Prior to the elimination of loser cells, such cells stain positive for phospho-JNK and phospho-PDH. [Redacted]

5. Is NR2 expression increased in all Myc expressing cells – or is it dependent on clonal expression of Myc?

No. NR2 levels are not increased in a homotypic Myc setting. Accordingly, NR2 levels are not elevated following tissue-wide expression of Myc (hh-Gal4>UAS-Myc) (new Fig. S8).

Suppl. Fig. S8

Is glutamate also increased in wing disc cells, or just in FB?

[Redacted]

This line of work will be pursued in a subsequent study.

6. On p. 8 it is stated, "...preventing loser cells from transferring lactate to their neighbours may remove fitness disparities and inhibit cell competition." Do wildtype cells take up lactate from each other during normal growth?

No. During normal growth, lactate does not seem to be exchanged.

To address this question, we expressed a genetically encoded intracellular lactate sensor in the wing pouch and generated cell clones in which the *Drosophila* lactate transporter MCT1 was downregulated by RNAi (new Fig. 4c) or pharmacologically inhibited (new Fig. 4d). In both settings (see red boxes) there were no significant differences in lactate levels between WT cells and control clones. This is in stark contrast to winner:loser relationships (new Fig. 4d, right panel).

d

new Fig. 4c,d: Wild-type cells to not take up lactate during normal growth. Lactate FRET signal was monitored in clones and in surrounding cells in the wing pouch (c,d). (d) Wing discs were cultured ex vivo and Lactate FRET signal measured over time following treatment with an MCT inhibitor

7. The increase in p-PDH in the NR2-depleted clones led to a decrease in mitochondrial size as assessed by Mitotracker red staining. Is a decrease in mitochondrial size important? Do mitochondria shrink with decreased activity?

Our new data demonstrate that NR2-depleted clones have mitochondria that **i**) have low TMRM staining (TMRM is a cell permeable, cationic, red-orange, fluorescent dye that is readily sequestered by active mitochondria), **ii**) are smaller in size, and **iii**) are positive for phospho-PDH (new Fig. 3a,b,c). Together, this is indicative of decreased mitochondrial activity.

Fig. 3a,b,c

8. Does activation of TNFR/JNK in clones of cells lead to an increase in p-PDH? The implication from their model is that it would. Or does this metabolic reprogramming only occur when NR2 is depleted?

Indeed, clonal over-expression of JNK triggers phosphorylation of PDH (new Fig. S5).

New Fig. S5

9. In Fig. 6A, B, the images of the DAPI channels here are oddly restricted to the periphery of the discs, making it difficult to look at the integrity of the disc. Could the authors provide better images of the DAPI channel? Also, the disc with Myc expressing clones in A shows a puzzling whirling morphology of the clones that is very unusual for Myc expressing clones. Is this representative of all their discs with Myc clone?

We have provided better images (new Fig. 8d).

10. Fig. 6F. The control (lacZ-RNAi) should be in Myc-expressing clones, not just GFP-expressing clones, in order to compare with the Myc-expressing, NR2-expressing clones.

We now provide the requested control. See **new Fig. 8d**, upper row.

New Fig. 8d

Minor/grammatical

→ We are thankful for highlighting all the minor/ grammatical issues, which we have amended in full.

1. The authors begin the paper by saying they carried out a candidate based approach to finding regulators of cell competition, however no other “candidates” are tested or mentioned. On the other hand, they present quite a detailed description of NMDA receptors and interactors by way of a rationale for studying NR2 - thus they don’t need to bring in the candidate approach as justification for their study.

We have followed the reviewer’s advice and modified our results section accordingly.

2. A reference for NR2:GFP is needed (Nagarkar-Jaiswal et al., 2015, eLife 4: e05338).

The suggested reference is now included.

3. On p. 6, why is the term “fitness disparity” in quotes and italics?

This has been corrected.

4. On p. 8, a typo: PHD (instead of PDH)

This has been corrected.

5. Fig. 3.

Panel A is a little misleading – the loser clone here is expressing NR2-RNAi, which leads to JNK activation in those cells, which according to their results is required to alter the cells’ metabolism. Or, as mentioned above, is Grnd and JNK activation sufficient to cause the metabolic programming?

We have clarified the scheme (**new Fig. 3f**). According to our model, in NR2-RNAi loser clones, Grnd-mediated activation of JNK results in phosphorylation of PDK, which in turn phosphorylates and inactivates PDH. This prevents the generation of Acetyl-CoA and its usage in the TCA, which ultimately causes metabolic reprogramming and lactate production.

6. I have issues with the authors’ use of the term “loser”, throughout the manuscript and Figures. For example, the authors state: “Importantly, RNAi-mediated depletion of NR2 or Dlg1 had no effect on clonal survival under homotypic condition (losers among losers), such as upon tissue-wide NR2 depletion using nubbin-Gal4 or hedgehog (hh)-Gal4”. While realizing that their descriptors (losers among losers and losers among winners) is an attempt to clarify, it actually makes things more confusing – It is important to remember that cells are only “losers” when in a heterotypic environment. In a homotypic environment, the cells are not “losers” and shouldn’t be called that.

We agree with this reviewer and have corrected the ms accordingly. Thus, only cells that are indeed eliminated in a heterotypic environment are marked as losers.

7. Several figure legends say “Loser clones are marked by GFP”, however, the control clones are also marked by GFP, and these are not “losers”. Some of the legends also say: “The indicated genes-of-interest (goi) were knocked down and over-expressed in GFP-marked clones”. This is more accurate - they don’t need to say ‘Loser’ clones.

We have corrected our ms accordingly.

Banretti and Meier Characterize the role of the NMDA receptor in cell competition.
 1-They show that signaling through this receptor mediates cell competition and
 2-that glucose/lactate metabolism and lactate transport plays a crucial role in NR2-mediated cell competition.
 3- [Redacted]

We thank the reviewer for the thoughtful comments. We have followed the advice and have considerably developed our NR2 story. We then tested the role of NR2 in Myc-mediated cell competition. [Redacted]

1-Regarding the NMDA story, there is no insight on how the activity of the channel (across which, cations can enter or exit the cell) impacts on cell fitness. In addition, it is not clear how the higher production of glutamate (intracellular, as judged by the images) impacts the pathway. Furthermore, it is not clear whether during cell competition winner cells activate NR2 or loser cells downregulate it and whether these expression changes are cell-autonomous or depending on the heterotypic context.

We thank this reviewer for raising these interesting questions. To address the role of the activity versus scaffolding function of NR2, we have used the NMDAR antagonist AP5 ((2R)-amino-5-phosphonopentanoate)¹³. Treatment with AP5 suppressed the loss of NR2 clones in a heterotypic genetic background (losers among winners) (Fig. S3a,b), phenocopying a homotypic setting. This illustrates that the competitive behaviour between NR2-losers and WT-winners is due to a relative difference in NR2 activity among competing clones. AP5-mediated global inhibition of NR2 thereby seems to eradicate the “fitness disparity” among competing clones.

Fig. S3a,b

Moreover, we have also tested the effect of AP5 (an NMDAR inhibitor) on Myc-mediated cell competition. Interestingly, we find that pharmacological inhibition of the activity of NMDAR suppresses the supercompetitor status of Myc (new Fig. 8g,h). These data are consistent with the notion that the activity of NMDAR mediates cell competition.

New Fig. 8g,h

Of note, we have removed the data regarding the glutamate levels in Myc supercompetitor clones. We agree that further work is needed to make firm conclusion about upregulation of glutamate in winner clones and how it influences NMDAR signalling in these cells. Interestingly, however, a similar autocrine glutamate-driven NMDAR-signalling circuit was identified in a mouse model of pancreatic neuroendocrine tumorigenesis (PNET) and in selected human cancers¹⁴.

2-Regarding the lactate transport hypothesis, while the authors clearly demonstrate the involvement of lactate production and transport for loser status acquisition, the proposed model does not explain the results. The authors propose that donation of Lactate from losers renders winner cells fitter because they obtain extra fuel. This idea is difficult to accept, provided that lactate transport is never against gradient and therefore lactate should always be more concentrated in loser cells (although this has not been directly measured) which, under the author's hypothesis would render them fitter. Furthermore, what triggers loser cell death under this hypothesis remains unknown. If it was JNK activation, then the whole Lactate story would not hold, as in principle it would work downstream JNK activation. Previous involvement of Warburg metabolism and Lactate production has been described in MDCK cells, so further elaboration would be needed here.

This is a key question that was also raised by Reviewer 1. See also our response at the beginning of this rebuttal letter.

Different tumours can behave differently. Some rely more on the Warburg effect while others depend on the reverse Warburg metabolism. This is highlighted by recent studies describing the metabolism of lactate in human lung tumours *in vivo* and experimental tumours in mice, showing that in certain tumours exogenous lactate is consumed and used as a respiratory fuel¹⁻³. Our *in vivo* data in flies are entirely consistent with these recent *in vivo* studies on the reverse Warburg metabolism. Clearly, this does not undermine the observations made using the MDCK cell line but merely indicates that our system differs and is in line with the above-mentioned studies.

To address the lactate transport question, we expressed a genetically encoded intracellular lactate sensor in the wing pouch and generated cell clones in which the *Drosophila* lactate transporter MCT1 was downregulated by RNAi (**new Fig. 4c**) or pharmacologically inhibited (**new Fig. 4d**) (attached below). We found that NR2-RNAi loser clones had significantly lower levels of intracellular lactate than surrounding wild-type cells or control clones (**new Fig. 4b,c**). Importantly, blocking lactate exchange via RNAi-mediated knockdown of *Mct1* in NR2-loser clones rescued lactate depletion in loser cells. Co-depletion of MCT1 not only prevented depletion of lactate in loser clones but also caused significant lactate accumulation (**new Fig. 4c**). No such changes in lactate levels were seen under non-competitive control conditions (**new Fig. 4c**).

Ex vivo treatment with an MCT inhibitor (MCTi) also led to a relative increase of lactate in losers, while it caused a corresponding decrease of lactate in surrounding winners (**new Fig. 4d**). Feeding L-lactic acid (LLA) to animals, like treating them with MCTi, suppressed lactate depletion in NR2-loser clones (**new Fig. 4e**). Together, these data are consistent with the notion that NR2-depleted loser cells are metabolically reprogrammed to produce and secrete lactate. See our **new data Fig. 4a,b,c,d,e**

Importantly, interference with this lactate transfer completely blocks cell competitive behaviour. See **previous Fig. 5a,b,c**.

Fig. 4c,d: NR2-loser cells transfer their lactate to winners. Lactate FRET signal was monitored in clones and in surrounding cells in the wing pouch. (d) Wing discs were cultured *ex vivo* and Lactate FRET signal measured over time following treatment with an MCT inhibitor.

Likewise, lactate-mediated metabolic coupling also seems to influence loser/Myc-supercompetitor relationships. Elevated intracellular lactate level was detected in Myc cells juxtaposed to WT surrounding cells (**new Fig. 6d**). Consistent with the notion that loser cells produce and transfer lactate to winners, we found that *ex vivo* treatment with an MCT inhibitor (MCTi), which blocks lactate exchange, led to a time-dependent increase of lactate in losers, while it caused a corresponding decrease of lactate in Myc winners (**new Fig. 6e**). This indicates that Myc supercompetitor cells receive and consume lactate from WT losers.

New Fig. 6d. Lactate-mediated metabolic coupling influences loser/Myc-supercompetitor relationships. Lactate FRET signal was monitored in clones and in surrounding cells in the wing pouch. Wing discs were cultured *ex vivo* and Lactate FRET signal measured over time following treatment with an MCT inhibitor.

→ Therefore, our data are entirely in line with the current literature on the reverse Warburg effect.

3-Regarding the effect of PDH/PDK, if PHD regulation is at the post-translational level, why does - PDH mRNA overexpression completely rescue loser clones and how is it that p-PDH is not detected in these clones?

This is an interesting question. Our new data demonstrate that clonal over-expression of PDH abrogates JNK signalling (**new Fig. S6b**), highlighting the presence of a regulatory feed-back loop.

New Fig. S6b

4-Regarding the Myc [Redacted]-NR2 connection, the authors demonstrate dominance of the loser phenotype induced by NR2 elimination over Myc [Redacted]. This is an interesting concept, especially in understanding and intervening tumor growth, however the NMDA pathway has already been described in this context. The observations reported here show that NMDA heterotypic loss leads to death despite Myc [Redacted] but this does not necessarily mean that differential NR2

activity mediates Myc [Redacted] cell competition, just dominance of one phenomenon versus the other.

Functional demonstration should involve showing that during Myc [Redacted] induced competition loser cells are eliminated due to differential NR2 activity. The easiest way to do this would be to induce Myc [Redacted] clones in a homotypic NR2-deficient background or in the presence of NMDA inhibitors. Such experiments would show whether inability to produce imbalance in NR2 activation between winner and loser cells blocks Myc [Redacted] induced competition. In the absence of these characterizations, the conclusions presented and the title do not clearly connect to the presented evidence.

We thank the reviewer for raising this important point. We have followed the Reviewer's advice and conducted a heterotypic Myc supercompetition assay in which NR2 was depleted tissue-wide.

Importantly, under conditions where NR2 was depleted tissue-wide, Myc clones no longer acquired supercompetitor status. Accordingly, wild-type cells (marked with GFP) were no longer eliminated by Myc clones (new Fig. 8e and f). Likewise, pharmacological inhibition of the activity of NMDAR suppresses the supercompetitor status of Myc (new Fig. 8g,h, point 1 above).

→ These data are consistent with the notion that NR2 contributes to the supercompetitor status of Myc clones.

New Fig. 8

Actually, several statements in the abstract do not correspond with any experiment presented in the manuscript, like “Pharmacological inhibition or genetic depletion of NR2 changes the supercompetitor status of oncogenic Myc [Redacted] clones into ‘superlosers’, resulting in their elimination via cell competition by wild-type neighbours in a TNF-dependent manner.” I have not seen in this manuscript any results involving TNF or NR2 pharmacological inhibition in Myc-mosaic studies.

We have corrected these statements.

5- [Redacted]

Minor point: The intro on NMDA receptors in the first part of the results could be moved to intro. We have shortened this introductory section.

Reviewer 3

Banreti and Meier present a very interesting and potentially significant report describing how the differential expression/activity of NMDA receptors in different cell populations in drosophila can decrease the survival of the weakly activated cell by promoting lactate shuttling to the competing cell. They show that the oncogenes myc [Redacted] appear to use this mechanism to out-compete normal cells.

Specific comments:

1) For the non-specialist, just how is the “Proportion of GFP+ clones / pouch” determined? Are these cells or cluster of cells? If cluster of cells, how are adjacent clusters distinguished from a larger cluster and how are the total number of clusters determined? If single cells, are these expressed as the fraction of DAPI+?

To quantify the proportion of GFP+ clones/ pouch, we used ImageJ software. First, an automatic thresholding method was used to select GFP positive areas. Then the region of interest (the entire wing pouch, which is the area where cell competition occurs within the wing disc) was defined and measured. Within this compartment (region of interest), the sum of GFP+ areas was calculated using the “Analyse Particles” command. The ratio of GFP+ clones / pouch was then simply determined by dividing the “sum of GFP+ area” with the “area of interest”.

2) Was the Proportion of GFP+ clones / pouch analysis performed blind to the conditions?

Yes. All experiments were done double-blind to the conditions. The same macros (described above) were used to analyse all clonal experiments.

3) Why did the authors pick the concentrations for the MCT-1 and NMDA receptor inhibitors that they did? The MCT-1 inhibitor is ~ 100 fold more potent at its target than the NMDA receptor inhibitor for its target, yet the MCT-1 inhibitor was used at a 20-fold higher concentration. Given that these inhibitors were provided in the food, the final target concentration would be very difficult to predict. Is there other literature or experimental evidence that these concentrations are sufficient?

We apologise for this mistake. We realised that there was a typo when referring to the concentration of the MCT inhibitor used. In the previous ms we indicated 100 μ M for the MCT inhibitor in the food when we really used 100 nM. For the *ex vivo* treatment of discs, 10 nM was used. We followed the guidance of ¹⁶. This mistake has been corrected.

4) Do the authors have any speculation as to how the differential activation of NMDA receptors in the two cell populations leads to JNK signaling?

This is an interesting question, which we are currently working on. JNK signalling is dependent on Grnd signalling. Therefore, it is possible that lower levels of NMDA receptor activity results in autocrine production of TNF, which in turn causes TNF/Eiger>Grnd-mediated activation of JNK. This in turn would drive activation of caspases as well as phosphorylation of PDK. PDK then phosphorylates and inactivates PDH, which shuts down mitochondrial function. Through a mechanism incompletely understood, this fuels JNK signalling, caspase activation and lactate export.

5) p. 5, the role of NMDA receptors in regulating lactate neuronal/astrocyte coupling is currently not widely accepted whereas the other actions are well established.

We have removed this section.

6) top of p. 8. The authors note that overexpressing PDH in NR2 loser clones prevented their elimination and prevented phospho-PDH staining. I follow why excess PDH may protect the cells, but why would phospho-PDH staining go down? If the loser clones with or without extra PDH had similar levels of active PDK, then one might expect the PDH/phospho-PDH ratio to change but the phospho-PDH levels might be the same (or higher, given more substrate)?

This is an interesting question. Our new data demonstrate that clonal over-expression of PDH abrogates JNK signalling (**new Fig. S6b**), highlighting the presence of a regulatory feed-back loop.

New Fig. S6b

7) typos: P9, line 2, supercompetitor; P9, line 5, need not capitalize "Glutamate P10, line 4, "may influences".

These have been corrected.

8) In the summary, "Myc cells upregulate NMDAR2 (NR2) to gain...". It might help some readers to specify that myc cells upregulate their NMDAR2 levels... since there is a possibility that myc cells somehow alter NMDAR2 levels in non-mutant cells.

This has been corrected.

References:

1. Faubert, B. *et al.* Lactate Metabolism in Human Lung Tumors. *Cell* **171**, 358-371 e359 (2017).
2. Hui, S. *et al.* Glucose feeds the TCA cycle via circulating lactate. *Nature* **551**, 115-118 (2017).
3. Harjes, U. Metabolism: More lactate, please. *Nat Rev Cancer* **17**, 707 (2017).
4. San-Millan, I. & Brooks, G.A. Reexamining cancer metabolism: lactate production for carcinogenesis could be the purpose and explanation of the Warburg Effect. *Carcinogenesis* **38**, 119-133 (2017).
5. Brooks, G.A., Dubouchaud, H., Brown, M., Sicurello, J.P. & Butz, C.E. Role of mitochondrial lactate dehydrogenase and lactate oxidation in the intracellular lactate shuttle. *Proc Natl Acad Sci U S A* **96**, 1129-1134 (1999).
6. Brooks, G.A. Lactate production under fully aerobic conditions: the lactate shuttle during rest and exercise. *Fed Proc* **45**, 2924-2929 (1986).
7. Cori, C.F. The Rate of Absorption of Epinephrine from the Subcutaneous Tissue. *Science* **70**, 355-356 (1929).
8. Chen, Y.J. *et al.* Lactate metabolism is associated with mammalian mitochondria. *Nat Chem Biol* **12**, 937-943 (2016).
9. Pavlides, S. *et al.* The reverse Warburg effect: aerobic glycolysis in cancer associated fibroblasts and the tumor stroma. *Cell Cycle* **8**, 3984-4001 (2009).
10. Sonveaux, P. *et al.* Targeting lactate-fueled respiration selectively kills hypoxic tumor cells in mice. *J Clin Invest* **118**, 3930-3942 (2008).

11. Bonucci, G. *et al.* The reverse Warburg effect: glycolysis inhibitors prevent the tumor promoting effects of caveolin-1 deficient cancer associated fibroblasts. *Cell Cycle* **9**, 1960-1971 (2010).
12. Bonucci, G. *et al.* Ketones and lactate "fuel" tumor growth and metastasis: Evidence that epithelial cancer cells use oxidative mitochondrial metabolism. *Cell Cycle* **9**, 3506-3514 (2010).
13. Olverman, H.J., Jones, A.W. & Watkins, J.C. L-glutamate has higher affinity than other amino acids for [3H]-D-AP5 binding sites in rat brain membranes. *Nature* **307**, 460-462 (1984).
14. Li, L. & Hanahan, D. Hijacking the neuronal NMDAR signaling circuit to promote tumor growth and invasion. *Cell* **153**, 86-100 (2013).
15. Strickaert, A. *et al.* Cancer heterogeneity is not compatible with one unique cancer cell metabolic map. *Oncogene* **36**, 2637-2642 (2017).
16. Payen, V.L. *et al.* Monocarboxylate Transporter MCT1 Promotes Tumor Metastasis Independently of Its Activity as a Lactate Transporter. *Cancer Res* **77**, 5591-5601 (2017).

Reviewers' Comments:

Reviewer #1:

Remarks to the Author:

I commend the authors for the additions and revisions to their very interesting manuscript, which are nicely described in an extensive and beautifully prepared rebuttal. Many of the concerns I noted in my previous review have been alleviated, making the data more convincing and the paper stronger. However, the new data raises some new concerns, which I have detailed below. Overall, I think the first part, that describes the experiments addressing the requirement for NR2 in for cell survival in mosaics – i.e., NR2's role in cell competition – is compelling and the data largely well done (but see suggestions below for clarification and additional quantification). What is still not convincing is the prominence of the role the authors ascribe to NR2 in Myc supercompetition. Here the data are not as compelling, less well controlled, and sometimes lack quantification. The authors make strong conclusions about NR2 and Myc that don't seem completely warranted by the data shown. The authors might consider toning down their conclusions with more circumspection.

Major points:

1. The authors make the claim that "NR2 over-expressing clones overgrew at the expense of wild-type surrounding cells". In Fig. 1h and i the GFP –positive area is certainly greater, and the 2 wing discs look similar in size in the images shown, supporting this view, but since in 1i the clonal area is normalized to wing pouch area, it is not certain. To confirm this, additional information is necessary about whether NR2-overexpression alters the timing of development or larval/disc size/growth rate. This is especially important since the *tub>CD2>Gal4* cassette will generate clones throughout the entire larva. Would the authors please provide the relevant information regarding this in the text?

2. In Fig. 2a and b, it appears from the dapi and phalloidin staining that tissue in the disc expressing NR2-RNAi is distorted, even outside of the clone area (compare with the controls above each). The MMP1 staining, which commonly marks sites of basement membrane disruption, is consistent with tissue distortion, but leads to the question of why does this occur? Is this a common feature of NR2-RNAi clones?

3. In Fig. 2c, the case is made that JNK signaling via Grnd, the Egr receptor, is activated cell autonomously in the NR2-RNAi clones. Where do the authors think that Egr is expressed? In the neighboring cells? In the clones themselves? (Or, is it expressed in hemocytes that are recruited to the tissue, as commonly seen with MMP1 induction?). Also in Fig. 2c, both UAS-p35 and UAS-Diap1 are used to suppress cell death. Was neither alone sufficient to block clone elimination? And, do the authors have any thoughts about why the effect of hep-RNAi or (especially) bsk-DN in the clones is so variable?

4. The authors state on p. 7 that "NR2-depleted clones appeared to have mitochondria that were less active and smaller in size." The authors propose that this is due to JNK activation leading to metabolic reprogramming. However, the cells are dying and or damaged. What do mitochondria look like (size and TMRM staining) and is PDH inactivated in cells in the homotypic background, where the NR2-depleted cells are not induced to die? And, if this metabolic reprogramming is particular to NR1-depleted cells in the process of being eliminated, can they rule out that it is not a general property of cells induced to die via JNK activation, but a specific reprogramming response to loss of NR2?

5. In Fig. 4b, it is not clear what is circled in the first panel. Is this a magnified view of the clones in the panels on the right? It is not clear what is being shown in merged images, nor how this relate to clones. Is the effect shown in 4c corrected by PDK-RNAi?

6. In Fig. 4d, the lactate fret signal goes up in clones, while it goes down in surrounding cells. How is this actually measured, taking into account the number of cells/cell area? How can signal

differences be measured in such small clones? Also, if winner cells take up lactate, shouldn't the sensor pick that up, e.g. in the surrounding WT cells in 4d?

7. In Fig. 5 it is shown that lactate feeding blocks elimination of NR2-depleted clones (does this suggest that the transport of lactate outside is what causes their death?). Likewise, knocking down Impl3 or MCT1 in nr2-i clones blocked their elimination. This suggests that if NR2-depleted cells cannot make lactate or transport it outside, they are not eliminated. They show one image of NR2-depleted cells with nuclear expression of MCT1. How often was this seen (please quantify)? Is this due to competition or does it also occur in a homotypic background? They conclude that "preventing loser cells from transferring lactate to their neighbours may remove fitness disparities and inhibit cell competition". This conclusion seems to be a bit strong for the data shown.

8. In Fig. 6a, it also looks like pPDH is induced within Myc-expressing clones as well as outside. How often is this observed? Only one image and an intensity plot of that image is shown; quantitative data is needed.

9. A key experiment is the following: "we found that ex vivo treatment with MCT inhibitors (MCTi), which blocks lactate exchange, led to a time-dependent increase of lactate in losers, while it caused a corresponding decrease of lactate in Myc winners (Fig. 6e)." It seems important to corroborate this with additional approaches in order to make the strong conclusion that "Myc supercompetitor cells receive and consume lactate from losers".

10. On p. 9 the authors state: "Myc clones seemed to have highly active mitochondria whose activity was dependent on NR2 (Fig. 6f)". This is based on TMRM staining, which measures the mitochondrial membrane potential, which is a function of efficient oxidative phosphorylation. The loss of TMRM staining in the NR2-i cells implies the opposite: that ATP synthase is defective (as occurs when it is killed by oligomycin). It has been reported (de la Cova et al 2014) that the mitochondria in Myc expressing supercompetitor cells, although more abundant, are defective in oxidative phosphorylation, which was proposed to force them to switch to a more glycolytic metabolic program. Although this was largely based on a cell culture model of Myc supercompetition where conditioned medium from mixed cultures was used to induce competition in naïve cell types, their data showed that Myc super-competitor cells make quite a bit of lactate, whereas the "loser" WT cells do not, seemingly the opposite of what is observed here. Arguably a cell culture model and an in vivo disc model are not equivalent, but in fact the culture model did seem to recapitulate all other known aspects of Myc supercompetition.

11. Several antibodies against NR2 (one against *Drosophila*, the others mammalian) were listed as used in the Methods section, but it is not clear which ones were used for the images in Fig. S1 or Fig. 7. Would the authors please be specific about which were used for each figure? Also, in figure S1B a GFP protein trap was used to monitor NR2 expression in a wing disc. The expression looks fairly strong compared to what is shown by antibody staining in Fig. 7A and scored in clones in 7B, which looks quite weak; and although there does seem to be a trend of higher signal intensity in the Myc clones in the graph, in many cases the differences don't look significant. How many discs were examined? Does the expression of the protein trap also increase in Myc-expressing clones? Were the authors able to verify the increase in Myc expressing cells by RT-pCR?

Minor points:

1. In the Methods: given that different conditions can yield females that are good (or bad) egg layers, how many eggs do 15 flies/vial lay under their culture conditions? Are the larvae uncrowded during the clone growth period, and do they develop at the same rate?

2. It is stated "After clone induction, flies were raised for 10 days at 24 °C. The numbers of eclosed imagoes and dead pupae were counted and the ratio of imago/total per vial was calculated." Did

the authors mean to say that larvae were raised for 10 days...?

3. Under the description of the homotypic competition assays, the authors call the control clones "nEGFP-positive 'pseudoclones'" – but in fact they are real (not 'pseudo') clones, just not competitive (neutral, or control clones?). Also, please note that in CyO the O is capitalized (O = Oster, a person's name).

4. The use of different markers to denote the FRT cassettes appears throughout the ms. and is confusing. FRTs are traditionally denoted by >, I recommend that be used throughout the ms. to conform to the literature. Moreover, sometimes |stop| is used - this is not correct and needs to be changed. In addition, the > should always be oriented in the same direction, typically > stop> (left to right, as in 5' 3'). This confusing variation occurs in the text, the methods, the lists of strains, and in the figures and should be corrected.

5. According to the Supp table of genotypes used, the flip-out cassette used for these experiments was tub>CD2>lexA,lexO-mCherry; lexO-LacZ/+, but on panel "d" of Figure 4 it is listed as act|CD2|LexA,lexO-mCherry, LexO>control (etc). Please correct to the proper genotype; also elsewhere if necessary (I happened to notice this and also a discrepancy in Fig.6, but did not go through the entire paper).

Reviewer #2:

Remarks to the Author:

The authors largely improved the manuscript by focussing on and substantially extending the experiments on the NMDR and lactate metabolism/transport in cell competition.

I have a few minor comments:

1- It does not seem appropriate to mention "disease" in the abstract, as disease has not been explored in this work

2- Similarly, the term "oncogenic" is used sometimes in the manuscript to refer to Myc-overexpressing cells. To my knowledge, Myc overexpressing cells do not produce tumors in this experimental setting, so I think it would be better not to use this term here (of course, authors may use it to discuss the potential implications of this study in different settings)

3- Mct1 is a main direct target of Myc in mammalian cells. If the authors have data on whether Myc regulates Mct1 in heterotypic or homotypic situations, it would be interesting to have these data included in the manuscript.

Miguel Torres

Reviewer #3:

Remarks to the Author:

The authors have made extensive revisions to an interesting paper. I do not have any remaining concerns.

Response to Reviewer comments on ms NCOMMS-18-14726B-Z

We would like to thank you and the reviewers for the additional comments and constructive criticisms. We were delighted to see that all three reviewers thought that our revised ms was exciting and well conducted. Nevertheless, Reviewer 1 raised additional concerns, which we have responded in full with additional data and explanations.

→ New key points:

1 (point 1). We provide additional data regarding whether NR2-overexpression alters the timing of development or larval/disc size/growth rate (our response to point 1). Our New Fig. 1j indicates that clonal over-expression of *NR2* does not alter developmental timing, organ or larval size (**New Fig. 1j**). Together, our data indicate that relative levels of NR2 underpins cell competitive behaviour in the wing epithelia.

2 (point 4). We demonstrate that PDH is not phosphorylated (inactivated) under homotypic settings. Accordingly, we find no pPDH staining following tissue-wide knockdown of *NR2* (**New Suppl. Fig. 5e**).

3 (point 8). To corroborate the reproducibility of the data shown in Fig. 6a we have included additional images and quantifications (**New Suppl. Fig. 8a,b**). These essentially show the same data. Moreover, our revised ms demonstrates that PDH is not phosphorylated in homotypic Myc settings (**New Suppl. Fig. 8c**).

4 (point 9). We have followed the reviewer's suggestion and have validated our finding that Myc supercompetitor depend on the transfer of lactate from losers. Accordingly, lactate feeding suppresses Myc-mediated supercompetition (**New Suppl. Fig. 8d,e**). Moreover, we find that pharmacological inhibition of MCT1 blocks Myc supercompetition (**New Suppl. Fig. 8d-g**). Together, these data corroborate the notion that uptake and consumption of lactate by Myc clones is required for their supercompetitor status.

5 (point 11). We are providing additional data, which corroborate the notion that the expression levels of NR2 correlate with the one of Myc (**New Fig. 7b-g**).

6. Moreover, we have provided additional explanations where further clarification was requested (points 2, 3, 5, 6, 7, 10 and minor points 1-5, and all points to reviewer 2).

Please find below our response to Reviewer 1 and 2, with their comments in blue, and our response in plain text. We have highlighted the respective text changes in the main ms in **yellow**.

Reviewer #1 (Remarks to the Author):

I commend the authors for the additions and revisions to their very interesting manuscript, which are nicely described in an extensive and beautifully prepared rebuttal. Many of the concerns I noted in my previous review have been alleviated, making the data more convincing and the paper stronger. However, the new data raises some new concerns, which I have detailed below. Overall, I think the first part, that describes the experiments addressing the requirement for NR2 in for cell survival in mosaics – i.e., NR2's role in cell competition – is compelling and the data largely well done (but see suggestions below for clarification and additional quantification). What is still not convincing is the prominence of the role the authors ascribe to NR2 in Myc supercompetition. Here the data are not as compelling, less well controlled, and sometimes lack quantification. The authors make strong conclusions about NR2 and Myc that don't seem completely warranted by the data shown.

The authors might consider toning down their conclusions with more circumspection.

Major points:

1. The authors make the claim that “NR2 over-expressing clones overgrew at the expense of wild-type surrounding cells”. In Fig. 1h and i the GFP –positive area is certainly greater, and the 2 wing discs look similar in size in the images shown, supporting this view, but since in 1i the clonal area is normalized to wing pouch area, it is not certain. To confirm this, additional information is necessary about whether NR2-overexpression alters the timing of development or larval/disc size/growth rate. This is especially important since the *tub>CD2>Gal4* cassette will generate clones throughout the entire larva. Would the authors please provide the relevant information regarding this in the text?

We would like to thank the reviewer for raising this important point. We found that clonal over-expression of NR2 did not alter developmental timing, organ or larval size (**new Fig. 1j**). We have amended our ms accordingly (highlighted in yellow).

→ We would like to emphasize that we used age-matched larvae for all clonal analysis throughout our study, including the one for control and the *UAS-NR2* (Fig. 1h and i). We have modified our ms to highlight this (**see first paragraph of 'Results' section**).

j NR2 overexpression - pouch size Heterotypic clonal analysis

New Fig. 1j. Clonal overexpression of NR2 does not alter developmental timing, organ or larval size.

2. In Fig. 2a and b, it appears from the dapi and phalloidin staining that tissue in the disc expressing NR2-RNAi is distorted, even outside of the clone area (compare with the controls above each). The MMP1 staining, which commonly marks sites of basement membrane disruption, is consistent with tissue distortion, but leads to the question of why does this occur? Is this a common feature of NR2-RNAi clones?

This is correct. We frequently see that clonal depletion of *NR2* (RNAi) can lead to distortions of the disc. Most likely this is due to the large amount of cell death (loser clone elimination) that occurs in these discs.

3. In Fig. 2c, the case is made that JNK signaling via Grnd, the Egr receptor, is activated cell autonomously in the NR2-RNAi clones. Where do the authors think that Egr is expressed? In the neighboring cells? In the clones themselves? (Or, is it expressed in hemocytes that are recruited to the tissue, as commonly seen with MMP1 induction?). Also in Fig. 2c, both UAS-p35 and UAS-Diap1 are used to suppress cell death. Was neither alone sufficient to block clone elimination? And, do the authors have any thoughts about why the effect of hep-RNAi or (especially) bsk-DN in the clones is so variable?

The source of Egr is an interesting subject, which is currently being investigated. We consider that this issue is beyond the scope of the present ms and is best addressed in a follow-up study.

We have not tested UAS-*p35* and UAS-*Diap1* alone. The variation of the data with *hep*-RNAi and *bsk*-DN is most likely due to differences in their expression levels.

4. The authors state on p. 7 that “NR2-depleted clones appeared to have mitochondria that were less active and smaller in size.” The authors propose that this is due to JNK activation leading to metabolic reprogramming. However, the cells are dying and/or damaged. What do mitochondria look like (size and TMRM staining) and is PDH inactivated in cells in the homotypic background, where the NR2-depleted cells are not induced to die? And, if this metabolic reprogramming is particular to NR2-depleted cells in the process of being eliminated, can they rule out that it is not a general property of cells induced to die via JNK activation, but a specific reprogramming response to loss of NR2?

Indeed, this is a JNK-mediated effect, as we have shown in Fig 4c.

Further, we find that PDH is not inactivated under homotypic settings. Accordingly, we find no pPDH staining following tissue-wide knockdown of NR2 (**New Fig. S5e**).

New Fig. S5e. NR2 controls cell competition by regulating lactate-mediated metabolic coupling between winners and losers. e, Homotypic analysis. Confocal images of anti-phospho-PDH stained wing pouches. Scale bar 50 μ m.

5. In Fig. 4b, it is not clear what is circled in the first panel. Is this a magnified view of the clones in the panels on the right? It is not clear what is being shown in merged images, nor how this relate to clones. Is the effect shown in 4c corrected by PDK-RNAi?

The encircled areas depict clones marked by RFP.

The first panels with the circles are magnifications of the area shown in the panels next to them. To make this much clearer, we have indicated the scale bars in all the panels. Scale bars are 20 μ m. The merged image shows an overlay of lactate FRET (green) and RFP (**New Fig. 4b**).

New Fig. 4b. Heterotypic clonal analysis. The indicated genes-of-interest (*goi*) were knocked down. Lactate FRET signal was monitored in clones and in surrounding cells in the wing pouch. Clones are outlined with dashed lines and marked by RFP (red). Scale bars 20 μ m.

Fig. 4c shows the relative FRET signal intensity, relative to control. It has not been corrected by *PDK-RNAi*.

6. In Fig. 4d, the lactate fret signal goes up in clones, while it goes down in surrounding cells. How is this actually measured, taking into account the number of cells/cell area? How can signal differences be measured in such small clones? Also, if winner cells take up lactate, shouldn't the sensor pick that up, e.g. in the surrounding WT cells in 4d?

This was measured by FRET using standard imaging software (ZEISS ZEN and ImageJ) as previously described [1]. The genetically encoded FRET sensor is highly sensitive and enables us to measure signal differences at single cell resolution. Surrounding WT cells not only take up lactate, they also consume it. Hence, the presence of lactate in winner clones is best visualised in a dynamic fashion over time, following inhibition of MCT1, as shown in Fig. 4d. Note, the graph shown in 4d is normalised to time 0. Hence, the amount of lactate in winners and losers is set to 1. This makes it appear as if Winners have the same amount of lactate as losers, which is not the case. Without MCT1 inhibitor treatment, surrounding WT cells have relatively lower FRET signal intensity compare to *Myc* cells, as shown on Fig. 6d.

1. Hudry, B., et al., *Sex Differences in Intestinal Carbohydrate Metabolism Promote Food Intake and Sperm Maturation*. Cell, 2019. **178**(4): p. 901-918 e16.

7. In Fig. 5 it is shown that lactate feeding blocks elimination of NR2-depleted clones (does this suggest that the transport of lactate outside is what causes their death?). Likewise, knocking down *Imp13* or *MCT1* in *nr2-i* clones blocked their elimination. This suggests that if NR2-depleted cells cannot make lactate or transport it outside, they are not eliminated. They show one image of NR2-depleted cells with nuclear expression of *MCT1*. How often was this seen (please quantify)? Is this due to competition or does it also occur in a homotypic background? They conclude that "preventing loser cells from transferring lactate to their neighbours may remove fitness disparities and inhibit cell competition". This conclusion seems to be a bit strong for the data shown.

The reviewer is correct. Our data indeed suggest that the transfer of lactate from losers to winners results in the death of loser cells. Thus, it appears that the death is due to the loss of carbon fuel in losers. However, it should be noted that there exists a complex feed-back mechanism between mitochondrial activity and JNK signalling, as outlined in our previous submission. This is evident as co-depletion of *PDK* fully rescued the elimination of *NR2*-loser clones (Fig. 3d,e) and abrogated the appearance of cleaved *DCP1* positive cells (Fig. S5e). Likewise, Gal4-driven expression of *PDH* in *NR2*-loser clones blocked their elimination (Fig. 3e and Fig. S6a). In both settings, surviving *NR2*-depleted clones were negative for anti-p-*PDH* staining (Fig. 3d and Fig. S6a). Moreover, local over-

expression of PDH abrogated JNK signalling (Fig. S6b). This highlights the presence of a complex feed-back regulatory loop.

In this revised version of our manuscript, we included quantification of our data and new representative images are shown in **New Suppl. Fig. 7b-e**. Shown on New Suppl. Fig. 7b, MCT1 staining is not nuclear.

We have changed the wording. We have removed the mentioning of 'fitness disparities'. We now state: "... preventing loser cells from transferring lactate to their neighbours inhibits cell competition."

New Suppl. Fig. 7b-e. MCT1 is upregulated in *NR2* loser clones.

b, Confocal images of anti-MCT1 stained wing pouches. Scale bars 20 μm (a) and 10 μm (b). See Supplementary Data Table for genotypes. **c**, Fluorescent intensities of anti-MCT1 (red) and GFP (green) are measured by ImageJ software at the yellow line. **d**, Pearson's correlation analysis of anti-MCT1 and GFP. **e**, Fluorescent intensities of anti-MCT1 are measured by ImageJ software. n depicts the number of cells analysed.

8. In Fig. 6a, it also looks like pPDH is induced within Myc-expressing clones as well as outside. How often is this observed? Only one image and an intensity plot of that image is shown; quantitative data is needed.

Low levels of pPDH is indeed present in Myc-expressing cells, but significantly higher levels of pPDH staining is observed in non-Myc cells (losers). To corroborate the reproducibility of the data shown in Fig. 6a we have included additional images and quantifications (**New Fig. S8a,b**). These essentially show the same data. Moreover, our revised ms demonstrates that PDH is not phosphorylated in homotypic Myc settings (**New Fig. S8c**).

New Suppl. Figure S8a, b and c. Lactate-mediated metabolic coupling occurs during loser/*Myc* supercompetition. **a**, Heterotypic clonal analysis. Confocal images of dissected wing pouches stained for anti-phospho-PDH (reading out its inactivation). Scale bar 25 μ m. **b**, Fluorescent intensities of phospho-PDH (black) and GFP (green) are measured by ImageJ software at the yellow lines. **c**, Homotypic clonal analysis. Confocal images of dissected wing pouches stained for anti-phospho-PDH (reading out its inactivation). Scale bar 25 μ m.

9. A key experiment is the following: “we found that ex vivo treatment with MCT inhibitors (MCTi), which blocks lactate exchange, led to a time-dependent increase of lactate in losers, while it caused a corresponding decrease of lactate in *Myc* winners (Fig. 6e).” It seems important to corroborate this with additional approaches in order to make the strong conclusion that “*Myc* supercompetitor cells receive and consume lactate from losers”.

We have followed the reviewer’s suggestion and have validated the above-mentioned finding using additional approaches. Accordingly, lactate feeding suppresses *Myc*-mediated supercompetition (**Suppl. Fig. S8d,e**). Moreover, we find that pharmacological inhibition of MCT1 blocks *Myc* supercompetition (**New Suppl. Fig. S8d-g**).

Together, these data corroborate the notion that uptake and consumption of lactate by *Myc* clones is required for their supercompetitor status.

New Suppl Fig. S8d,e,f,g. **d**, Heterotypic clonal analysis. Myc was over-expressed in *GFP*-marked clones. For genotypes see Supplementary Data Table. Clones are marked by *GFP* (green). Lactate feeding and MCTi treatment was conducted as outlined in the Methods section. Scale bar 100 μ m. **e**, Quantification of the Lactate (LLA) and MCTi treatment assays. **f**) Myc supercompetition assay. WT cells are marked with *mCherry*. Treatment with MCT1 inhibitor was conducted as outlined in the Methods section. Scale bar 100 μ m. **g**, Quantification of the MCTi treatment assay. See Supplementary Data Table for genotypes. Error bars represent average occupancy of the indicated UAS clones per wing pouch \pm SD. **** P <0.0001, *** P <0.001, ** P <0.01, * P <0.1 by Mann-Whitney nonparametric *U*-test. *n* depicts the number of wing discs. See Supplementary Data Table for genotypes.

10. On p. 9 the authors state: "Myc clones seemed to have highly active mitochondria whose activity was dependent on NR2 (Fig. 6f)". This is based on TMRM staining, which measures the mitochondrial membrane potential, which is a function of efficient oxidative phosphorylation. The loss of TMRM staining in the NR2-i cells implies the opposite: that ATP synthase is defective (as occurs when it is killed by oligomycin). It has been reported (de la Cova et al 2014) that the mitochondria in Myc expressing supercompetitor cells, although more abundant, are defective in oxidative phosphorylation, which was proposed to force them to switch to a more glycolytic metabolic program. Although this was largely based on a cell culture model of Myc supercompetition where conditioned medium from mixed cultures was used to induce competition in naïve cell types, their data showed that Myc super-competitor cells make quite a bit of lactate, whereas the "loser" WT cells do not, seemingly the opposite of what is observed here. Arguably a cell culture model and an *in vivo* disc model are not equivalent, but in fact the culture model did seem to recapitulate all other known aspects of Myc supercompetition.

Drosophila S2 cells are generally grown in the presence of **2000 mg/L glucose** in the culture medium (Schneider's Insect medium). Most cells grown in glucose-containing medium generate almost all their ATP via glycolysis despite abundant oxygen supply and functional mitochondria, a phenomenon known as the **Crabtree effect**. By contrast, most cells within the body rely on mitochondrial oxidative phosphorylation (OXPHOS) to generate the bulk of their energy supply.

Therefore, we agree that cell culture models, in which cells are grown in 2 mM glucose, are not equivalent to the *in vivo* setting. To our knowledge, *de la Cova et al.* did not show that Myc cells indeed produce lactate. They showed that the intracellular lactate level in Myc cells is higher when such cells are grown in 2 mM glucose, which is consistent with the Crabtree effect (it is worth to look at this article: A.I. Mot et al., Int J Biochem Cell Biol. 2016 Oct;79:128-138).

While it is clear that cell culture models and an *in vivo* disc model are not equivalent, it is important to point out that our data do not contradict *de la Cova et al.* (2014) because enhanced glucose uptake could also occur. Many tumour cells can consume both glucose as well as lactate, and can readily switch between the two carbon sources. See also A.I. Mot et al., *Int J Biochem Cell Biol.* 2016 Oct;79:128-138.

11. Several antibodies against NR2 (one against *Drosophila*, the others mammalian) were listed as used in the Methods section, but it is not clear which ones were used for the images in Fig. S1 or Fig. 7. Would the authors please be specific about which were used for each figure? Also, in figure S1B a GFP protein trap was used to monitor NR2 expression in a wing disc. The expression looks fairly strong compared to what is shown by antibody staining in Fig. 7A and scored in clones in 7B, which looks quite weak; and although there does seem to be a trend of higher signal intensity in the Myc clones in the graph, in many cases the differences don't look significant. How many discs were examined? Does the expression of the protein trap also increase in Myc-expressing clones? Were the authors able to verify the increase in Myc expressing cells by RT-pCR?

We apologise for this oversight. We have corrected the ms and now clearly specify the respective antibodies used.

We are providing additional data, which corroborate the notion that the expression levels of NR2 correlate with the one of Myc (See New Fig. 7b-e). Furthermore, endogenous variation in the expression levels of NR2 (monitored by NR2-Gal4; UAS-GFP) showed a significant correlation with Myc expression levels (anti-Myc immunostaining) (See New Fig 7f-g). The upregulation of NR2 was competition-dependent, as in a homotypic context, no difference in NR2 expression was observed between Myc expressing cells (driven with *hh*-Gal4) and wild-type cells Fig. S9.

New Fig 7b-g. Immunostaining of imaginal wing discs and fat body, GFP-marked Myc overexpressing clones. NR2 expression is revealed by antibody specific to the *Drosophila* NR2 subunit. Scale bars 100 µm and 25 µm, respectively c, Fluorescent intensities of anti-NR2 (red) and GFP (*Myc*, green) are measured by ImageJ software at the yellow lines. d, Pearson's correlation

analysis of anti-NR2 and GFP (Myc). e, Mean signal intensities of anti-NR2 specific staining in Myc expressing cells (blue) and immediately adjacent wild type cells (red). f, Endogenous expression levels of NR2 (followed by GFP) and Myc (anti-Myc immunostaining). Scale bar 20 μ m. d, Pearson's correlation analysis of NR2 (GFP) and Myc (anti-Myc).

[Redacted]

Minor points:

1. In the Methods: given that different conditions can yield females that are good (or bad) egg layers, how many eggs do 15 flies/vial lay under their culture conditions? Are the larvae uncrowded during the clone growth period, and do they develop at the same rate?

15 females lay about 30 eggs in 3 hrs. This ensures that our larvae develop in uncrowded settings and are harvested at specific time points during their development. Under these conditions, all animals develop at the same rate.

2. It is stated "After clone induction, flies were raised for 10 days at 24 °C. The numbers of eclosed imagoes and dead pupae were counted and the ratio of imago/total per vial was calculated." Did the authors mean to say that larvae were raised for 10 days...?

[Redacted]

3. Under the description of the homotypic competition assays, the authors call the control clones "nEGFP-positive 'pseudoclones'" – but in fact they are real (not 'pseudo') clones, just not competitive (neutral, or control clones?). Also, please note that in CyO the O is capitalized (O = Oster, a person's name).

We have changed "pseudoclones" to "non-competitive clones", and "Cyo" to "CyO".

4. The use of different markers to denote the FRT cassettes appears throughout the ms. and is confusing. FRTs are traditionally denoted by >, I recommend that be used throughout the ms. to conform to the literature. Moreover, sometimes |stop| is used - this is not correct and needs to be changed. In addition, the > should always be oriented in the same direction, typically > stop> (left to right, as in 5' 3'). This confusing variation occurs in the text, the methods, the lists of strains, and in the figures and should be corrected.

We would like to thank for highlighting this mistake. We have corrected it in our revised manuscript.

5. According to the Supp table of genotypes used, the flip-out cassette used for these experiments was $tub>CD2>lexA,lexO-mCherry; lexO-LacZ/+$, but on panel "d" of Figure 4 it is listed as $act|CD2|LexA,lexO-mCherry, LexO>control$ (etc). Please correct to the proper genotype; also elsewhere if necessary (I happened to notice this and also a discrepancy in Fig.6, but did not go through the entire paper).

We have corrected this mistake. The genotype of Figure 4d is: $yw; tub>CD2>lexA, lexO-mCherry; lexO-LacZ/+$

Reviewer #2 (Remarks to the Author):

The authors largely improved the manuscript by focussing on and substantially extending the experiments on the NMDR and lactate metabolism/transport in cell competition.

I have a few minor comments:

1- It does not seem appropriate to mention "disease" in the abstract, as disease has not been explored in this work

We agree with the reviewer and have removed the word 'disease'.

2- Similarly, the term "oncogenic" is used sometimes in the manuscript to refer to Myc-overexpressing cells. To my knowledge, Myc overexpressing cells do not produce tumors in this experimental setting, so I think it would be better not to use this term here (of course, authors may use it to discuss the potential implications of this study in different settings)

We have followed the reviewer's suggestion and have removed the term 'oncogenic'.

3- Mct1 is a main direct target of Myc in mammalian cells. If the authors have data on whether Myc regulates Mct1 in heterotypic or homotypic situations, it would be interesting to have these data included in the manuscript.

This is indeed an interesting point, which we will explore further in the future. Unfortunately, at present we do not have such data.

Reviewers' Comments:

Reviewer #1:

Remarks to the Author:

I thank the authors for nicely addressing each of my questions and carrying out additional supportive experiments. In the two new experiments lactate is fed to larvae or wing discs are treated ex vivo with an inhibitor of MCT1 (S. Fig 8d, e). In each experiment both the winners and losers get the treatment, making interpretations complicated. However, given all of the results, the data do support the idea that NR2 contributes to the Myc supercompetitor status via lactate exchange. In addition, the authors answered many of my other concerns, thus the paper is almost there! Only a few things remain, mostly explanations and simple fixes of text etc, detailed below.

1. Thank you for adding the wing pouch size data to Fig. 1j and corresponding text. Since in the text it is noted that "Of note, clonal over-expression of NR2 did not alter developmental timing, organ or larval size (Fig. 1j)" but only pouch size is shown. 'Age-matched' does not necessarily mean larvae or discs are developmentally matched, please either show the developmental timing and larval size data, or state 'data not shown' so it is clear that these parameters were also examined.

2. On page 5, lines 102-103 the following sentence appears: "clonal knockdown of *dlg1* or *scribble* (*scrib*) resulted in the elimination of 'loser' knockdown-clones...."
This sentence doesn't make grammatical sense –

3. Would the authors please add to the text a statement about seeing frequent tissue distortions when NR2 is clonally depleted? Tissue distortion is actually not a common feature of other contexts of "losers", so I think it is worth mentioning. [Perhaps it reflects something about loss of normal NR2 interactions with a binding partner (*Dlg*?) involved in cell-cell contacts?]

4. Reviewer 2 asked (and I concur) that the term "oncogenic" be removed in discussions of Myc supercompetition, yet the word still remains in the text in several places (e.g., abstract, twice on p. 9, and once on p10). As pointed out, all evidence indicates that myc-expressing (supercompetitor) clones are not tumorigenic in this experimental setting. I understand the authors' desire to draw analogies between cancer and Myc super-competition (and many in the field agree with this view), but using the term oncogenic just is not appropriate here.

5. Related to the point above, Myc-expressing clones are not "hyperplastic" (p. 9). Such clones do not proliferate faster than control (GFP) clones, but they are larger due to an increase in cell size (Johnston et al 1999). They also do not make the tissue overgrow (as say, a *Yki*-expressing clone does).

6. In the abstract the word "superlosers" is used. What is superloser supposed to mean? It seems gimmicky rather than informative.

7. Related to point 5: in the Methods it is stated that clones are induced at 72hr AEL, and allowed to grow for either 48 or 72 hrs. This means that the discs were fixed and examined at either 120hr AEL (at 25°C, late wandering or WPP) or 144hr AEL (well into pupal development, with discs starting to evert). Exactly how long were the clones in each figure allowed to grow? I suspect that the description in the Methods is in error (maybe they meant to say 24 or 48 hrs?).

8. In the experiment (Fig. 8e) in which they prevent elimination of WT loser clones with homotypic expression of NR2-RNAi, the image looks like there are many clones outside of the Nubbin domain (i.e., in the ventral hinge region in the panel). This seems odd, since in the middle panel, showing that all WT clones are eliminated (WP and hinge included). Since Nubbin-lexA presumably reflects the normal Nubbin domain, it wouldn't be expected that clones outside of the domain would be

rescued. Would the authors explain this?

9.The image of anti-DCP1 in Fig. 8d is very dim and difficult to see even when enlarged on a computer.

10.Why in the NR2-RNAi clones in S.Fig. 7b is MCT1 protein, by anti-MCT1 immuno-staining, completely nuclear? This seems very odd.

11.Also in that figure, the legend says "wing pouches"; as there is only one shown pouch should be singular.

12.In the Rebuttal it was stated that the respective NR2 antibodies would be clearly specified, but I cannot see where this information is. Were the same antibodies used for Fig. 7a and for S. Fig9?

13.Also for S.Fig 9, although the antibodies are weak, it looks to me like staining in the Anterior cells (lower panel of NR2 staining) of the discs look slightly increased relative to the posterior, where Myc is expressed (at fairly high levels with the HhGal4 driver). There is a sharp border in the staining at the A/P boundary (compare to the image above). If so, does Myc actually downregulate NR2 expression in a (non-competitive) "homotypic" environment..?

14.In the Methods section under Fly strains, a citation is needed for the tub>CD2>LexA and tub>myc>LexA flies (Alpar et al, Dev Cell 2018).

15.In figure 8f the text says "Myc supercompetition assay", and the genotypes in the Supp Table say they are "hsFlp;tub>myc y+>Gal4, UAS-GFP/nub-LexA; lexO-NR2-RNAi/+". However, the X-axis legends says "Myc clones". Shouldn't this be WT clones, since they are clones made in the tub>myc, y+>Gal4 background?

16.The authors make the statement that "NMDA receptor is responsible for fitness surveillance" in the Discussion. What do they mean by fitness surveillance? I'm not sure I understand where this conclusion comes from. More concretely, their data show that NR2 is required in wing disc cells for their ability to compete in mosaics with WT cells.

Response to Reviewer comments on ms NCOMMS-18-14726C

We would like to thank you for the positive decision on our manuscript. Below are our responses to the minor issues raised by Reviewer 1. We have responded to them in full with additional text changes.

Reviewer 1's comments are in blue while our response is in plain text. We have highlighted the respective text changes in the manuscript in yellow.

Reviewer #1 (Remarks to the Author):

1. Thank you for adding the wing pouch size data to Fig. 1j and corresponding text. Since in the text it is noted that "Of note, clonal over-expression of NR2 did not alter developmental timing, organ or larval size (Fig. 1j)" but only pouch size is shown. 'Age-matched' does not necessarily mean larvae or discs are developmentally matched, please either show the developmental timing and larval size data, or state 'data not shown' so it is clear that these parameters were also examined.

We refer to these data as 'data not shown'.

2. On page 5, lines 102-103 the following sentence appears: "clonal knockdown of dlg1 or scribble (scrib) resulted in the elimination of 'loser' knockdown-clones..."

This sentence doesn't make grammatical sense –

This was corrected.

3. Would the authors please add to the text a statement about seeing frequent tissue distortions when NR2 is clonally depleted? Tissue distortion is actually not a common feature of other contexts of "losers", so I think it is worth mentioning. [Perhaps it reflects something about loss of normal NR2 interactions with a binding partner (Dlg?) involved in cell-cell contacts?]

We have added this.

4. Reviewer 2 asked (and I concur) that the term "oncogenic" be removed in discussions of Myc supercompetition, yet the word still remains in the text in several places (e.g., abstract, twice on p. 9, and once on p10). As pointed out, all evidence indicates that myc-expressing (supercompetitor) clones are not tumorigenic in this experimental setting. I understand the authors' desire to draw analogies between cancer and Myc super-competition (and many in the field agree with this view), but using the term oncogenic just is not appropriate here.

We have removed the word 'oncogenic'.

5. Related to the point above, Myc-expressing clones are not "hyperplastic" (p. 9). Such clones do not proliferate faster than control (GFP) clones, but they are larger due to an increase in cell size (Johnston et al 1999). They also do not make the tissue overgrow (as say, a Yki-expressing clone does).

We have removed the word 'hyperplastic'.

6. In the abstract the word "superlosers" is used. What is superloser supposed to mean? It seems gimmicky rather than informative.

We respectfully disagree. We are using the term superloser to indicate that a supercompetitor has lost its competitive advantage and is eliminated.

7. Related to point 5: in the Methods it is stated that clones are induced at 72hr AEL, and allowed to grow for either 48 or 72 hrs. This means that the discs were fixed and examined at either 120hr AEL (at 25°C, late wandering or WPP) or 144hr AEL (well into pupal development, with discs starting to evert). Exactly how long were the clones in each figure allowed to grow? I suspect that the description in the Methods is in error (maybe they meant to say 24 or 48 hrs?).

We thank the reviewer for spotting this error, which has been corrected and clarified. Clones were induced 48 hrs after egg laying. Dissections were performed 72 hrs following clone induction, except for the *ex vivo* Lactate FRET experiments where wing discs were dissected 48 hrs after clone induction. We have modified the Materials and Methods section accordingly.

8. In the experiment (Fig. 8e) in which they prevent elimination of WT loser clones with homotypic expression of NR2-RNAi, the image looks like there are many clones outside of the Nubbin domain (i.e., in the ventral hinge region in the panel). This seems odd, since in the middle panel, showing that all WT clones are eliminated (WP and hinge included). Since Nubbin-lexA presumably reflects the normal Nubbin domain, it wouldn't be expected that clones outside of the domain would be rescued. Would the authors explain this?

The early developmental expression of nubbin is broader than the one at late L3 wandering stage. Nubbin expresses in the hinge at earlier developmental stages, prior to the occurrence of cell competition. The expression of nubbin is very defined at later stages, at the time when cell competition occurs, but not at earlier time points.

See " Ref <https://dev.biologists.org/content/develop/121/2/589.full.pdf>, *Although nubbin is expressed throughout the wing primordium, analysis of genetic mosaics suggests a localized requirement for nubbin activity in the wing hinge.*"

Since we are using RNAi to deplete NR2, the effect can last for several days, even though by that time the nub expression pattern is different.

9. The image of anti-DCP1 in Fig. 8d is very dim and difficult to see even when enlarged on a computer.

We have improved the image.

10. Why in the NR2-RNAi clones in S. Fig. 7b is MCT1 protein, by anti-MCT1 immuno-staining, completely nuclear? This seems very odd.

As shown in Supplementary Fig. 7c, anti-MCT1 staining is not nuclear, the signal is partially overlapping with the GFPnls signal but broader than that. In fact, the cells are dying and rounded up with a very tiny cytoplasmic fraction, as is frequently observed with apoptotic cells. At this resolution, particularly in the case of apoptotic cells it is very difficult to make a conclusion of the subcellular localization of MCT1, our intention was only to address the relative difference in the expression level of NR2-RNAi clones and that of the surroundings.

11. Also in that figure, the legend says "wing pouches"; as there is only one shown pouch should be singular.

The figure legends states: "a,b, Confocal images of anti-MCT1 stained wing pouches." this refers to the wing pouches shown in a and b, respectively.

12. In the Rebuttal it was stated that the respective NR2 antibodies would be clearly specified, but I cannot see where this information is. Were the same antibodies used for Fig. 7a and for S. Fig9?

We have used only one anti-NR2 antibody, which is clearly indicated in the Materials and Methods section.

13. Also for S. Fig 9, although the antibodies are weak, it looks to me like staining in the Anterior cells (lower panel of NR2 staining) of the discs look slightly increased relative to the posterior, where Myc is expressed (at fairly high levels with the HhGal4 driver). There is a sharp border in the staining at the A/P boundary (compare to the image above). If so, does Myc actually downregulate NR2 expression in a (non-competitive) "homotypic" environment..?

We have not evaluated this aspect.

14. In the Methods section under Fly strains, a citation is needed for the tub>CD2>LexA and tub>myc>LexA flies (Alpar et al, Dev Cell 2018).

This has now been included.

15. In figure 8f the text says "Myc supercompetition assay", and the genotypes in the Supp Table say they are "hsFlp;tub>myc y+>Gal4, UAS-GFP/nub-LexA; lexO-NR2-RNAi/+". However, the X-axis

legends says “Myc clones”. Shouldn’t this be WT clones, since they are clones made in the tub>myc, y+>Gal4 background?

We thank the reviewer for spotting this error. Indeed, this should read WT clones. This has been corrected.

16. The authors make the statement that “NMDA receptor is responsible for fitness surveillance” in the Discussion. What do they mean by fitness surveillance? I’m not sure I understand where this conclusion comes from. More concretely, their data show that NR2 is required in wing disc cells for their ability to compete in mosaics with WT cells.

Our data support the notion that the level of NMDA receptor determines cellular fitness. Lower levels result in cell elimination, while higher levels boost winner status. Therefore, it is likely that changes in NMDA receptor levels and activity leads to surveillance of tissue fitness. This is an attractive model that is worth discussing. Hence, this speculative viewpoint is discussed in the final section.